# Mutational profiling of SARS-CoV-2 papain-like protease reveals requirements for function, structure, and drug escape

Xinyu Wu [1,2], Margareta Go [1], Julie V. Nguyen [1], Nathan W. Kuchel[1,2], Bernadine G. C. Lu [1,2], Kathleen Zeglinski [1,2], Kym N. Lowes [1,2], Dale J. Calleja [1,2], Jeffrey P. Mitchell [1,2], Guillaume Lessene [1,2,3], David Komander [1,2], Matthew E. Call [1,2] & Melissa J. Call [1,2] ✉

Papain-like protease (PLpro) is an attractive drug target for SARS-CoV-2 because it is essential for viral replication, cleaving viral poly-proteins pp1a and pp1ab, and has de-ubiquitylation and de-ISGylation activities, affecting innate immune responses. We employ Deep Mutational Scanning to evaluate the mutational effects on PLpro enzymatic activity and protein stability in mammalian cells. We confirm features of the active site and identify mutations in neighboring residues that alter activity. We characterize residues responsible for substrate binding and demonstrate that although residues in the blocking loop are remarkably tolerant to mutation, blocking loop flexibility is important for function. We additionally find a connected network of mutations affecting activity that extends far from the active site. We leverage our library to identify drug-escape variants to a common PLpro inhibitor scaffold and predict that plasticity in both the S4 pocket and blocking loop sequence should be considered during the drug design process.

COVID-19, caused by SARS-CoV-2 has swiftly become one of the leading causes of death globally and, despite robust population-level immunity, remains so four years after SARS-CoV-2 emerged in the human population. SARS-CoV-2 can be treated by inhibition of crucial viral proteases. SARS-CoV-2, like SARS-CoV and other coronaviruses, relies on the Main protease (Mpro) and Papain-Like protease (PLpro) to cleave virally encoded poly-protein chains that contain non-structural proteins required for viral replication. Development of Pfizer's SARS-CoV-2 Mpro inhibitor, Nirmatrelvir[1] relied on lead compounds to an evolutionarily related coronavirus, SARS-CoV, which emerged in 2002. Further development was abandoned when SARS-CoV virus was eliminated in the human population but resumed in 2020 during the COVID-19 pandemic. Nirmatrelvir demonstrates that protease inhibition is effective in SARS-CoV-2 treatment, but viral escape from existing single-agent therapeutics remains a concern.

Based on recent findings and knowledge of SARS-CoV PLpro function, PLpro is an active enzymatic domain of non-structural protein 3 (Nsp3) (Fig. 1a). Like Mpro, PLpro is an attractive target for inhibitor development because of its proteolytic role in releasing Nsp1, Nsp2, Nsp3, and in conjunction with Mpro, Nsp4 from the viral poly-proteins pp1a and pp1ab[2]. Additionally, PLpro efficiently removes the post-translational modifications of ubiquitin and interferon-stimulated gene 15 (ISG15) from viral and host proteins[3–6], a mechanism that likely protects viral proteins[7] and hampers immune signaling pathways[8–12]. Proteolytic cleavage, de-ubiquitination and de-ISGylation activity relies on the motif common among all substrates, $Leu_{P4}X_{P3}Gly_{P2}Gly_{P1}\downarrow$ where X is commonly Lys or Arg and the arrow represents the site of cleavage[13].

The PLpro core has a domain architecture that resembles human de-ubiquitinases from the USP family and comprises "thumb", "palm" and "fingers" domains. The active site residues are found on a flat

[1]The Walter and Eliza Hall Institute of Medical Research, Parkville, VIC, Australia. [2]Department of Medical Biology, University of Melbourne, Parkville, VIC, Australia. [3]Department of Biochemistry and Pharmacology, University of Melbourne, Parkville, VIC, Australia. ✉e-mail: mjcall@wehi.edu.au

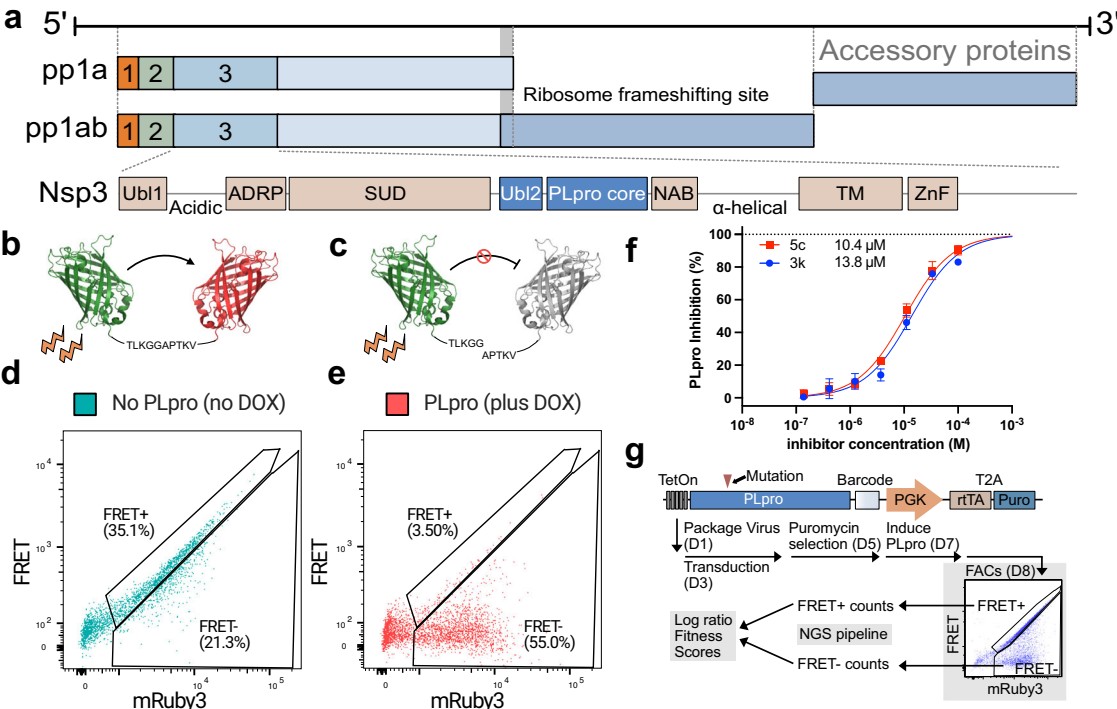

**Fig. 1 | SARS-CoV-2 genome and cellular assays for PLpro activity. a** SARS-CoV-2 genome indicating position of PLpro. Numbered domains 1, 2 and 3 refer to the first three non-structural proteins that are released by PLpro. Ubiquitin-like domain 1 (Ubl1), ADP-ribose phosphatase (ADRP), SARS-unique domain (SUD), ubiquitin-like domain 2 (Ubl2), Papain-like protease (PLpro core), nucleic acid binding domain (NAB), transmembrane domain (TM) and zinc finger motif (ZnF)[76] **b** FRET biosensor. mClover3 (green) covalently linked to mRuby3 (red) with the linker sequence TLKGGAPTKV. **c** Cleavage of the linker sequence by PLpro releases mRuby3 from mClover3 donor fluorophore, reducing FRET. **d** Flow cytometry plot of cells (gated on forward and side scatter as in SupplementaryFig. 1ai) containing the FRET biosensor prior to induction of PLpro expression with dox. The x-axis shows emission from the 610/20 filter after excitation with the 561 nm laser (mRuby3), the y-axis shows 610/20 emission after excitation with the 488 nm laser (FRET). **e** The same cells in (**d**) 24 h after dox addition, showing cleavage of the

FRET biosensor. **f** Dose response of 3k and 5c in the cellular assay described in **d**) & **e**). Data were normalized so the top was 100 and bottom was 0 after fitting the 5c dose response curve that had a Hill slope of 1. EC50 values for this representative experiment were calculated as: 13.8 μM (3k); 10.4 μM (5c), $n = 3$ and error bars are mean ± SD. Note: the 5c EC50 over 8 biological replicates is 15.5 ± 1.3 SEM (Supplementary Fig. 5c) **g** Schematic of Activity DMS workflow. The linear vector map shows the PLpro expression construct with Teton promoter element driving PLpro expression. A 16 bp barcode is installed 3′ of the PLpro coding sequence. This is followed by a constitutive PGK promoter that drives rtTA and Puromycin resistance. The workflow describes the PLpro Activity DMS assay, starting from viral packaging, reporter cell transduction with the library, dox induction, selection by FACS, Next-Generation Sequencing (NGS) and data processing. Source data are provided as a Source Data file.

surface, split among the thumb (Cys111) and palm (His272 and Asp286) domains, while the fingers domain contains a zinc binding site coordinated by Cys189, Cys192, Cys224, and Cys226 that is required for overall structural integrity[3]. Because the P1 and P2 positions of the cleavage motif are glycines, entry to the active site is narrow and the active site itself is shallow, making it spatially confined. Most well-characterized inhibitors of SARS-CoV-2 PLpro instead target the S4 pocket that accommodates the P4 Leu of the recognition sequence motif[13] and engage a flexible β-hairpin, termed the blocking loop, that flips up to cover the pocket[14–16].

Known inhibitors are focused on a few primary scaffolds exemplified by GRL0617[17], and the 5c/3k series[18], which were both developed from hits in high-throughput screens against SARS-CoV PLpro. GRL0617 and 5c/3k share a naphthyl ring that binds the S4 pocket, with polar contacts further stabilizing binding through Asp164 and a substituted phenyl ring that induces a conformation change in Leu162 allowing the compounds to be accommodated[14]. Recently reported compounds XR8-89 and Jun9-84-3 build on the GRL0617 scaffold. XR8-89 by replacing the naphthalene with a substituted phenylthiophene, and introducing the azetidine group to bind with Glu167[19] and Jun9-84-3 by replacing the phenyl ring with an indole ring, increasing potency[20]. Modifications have also been performed on 5c and 3k scaffolds to optimize absorption, distribution, metabolism and excretion (ADME) properties[21].

In this work, we establish an assay to measure PLpro activity in mammalian cells and construct a Deep Mutational Scanning (DMS)[22] library to assess the impact of almost all possible single-site substitutions on proteolytic activity. We further examine the mutational effects on protein stability by linking a fluorescent reporter to PLpro to measure the abundance of each variant in cells, allowing variants that impact PLpro activity, but not its folding, to be elucidated. We also use our library to predict variants of PLpro that escape the PLpro inhibitor scaffold exemplified by small molecule compounds 3k and 5c[6,18,21]. Top-ranked escape variants are validated individually in cellular assays and recombinant enzymatic assays. In combination, these data elucidate all regions of PLpro that are required for activity and indicate a role for PLpro flexibility in proteolytic cleavage and drug escape.

## Results

### Development of cell-based assay to measure PLpro activity

Deep mutational scanning potentially examines the functional effects of every single-site substitution in a region of interest of a protein. We chose to examine 315 residues of Nsp3 spanning residues 746 to 1060, which encompasses the PLpro core and the ubiquitin-like domain N-terminal to PLpro. This region can be expressed in isolation, remaining enzymatically active and in the remainder of the paper, we use PLpro numbering (1 to 315). Within our construct there are, including wildtype, 6301 theoretical protein variants, numbers which

make individual assessment of each variant a formidable task. Instead, we used DMS to assess the function of each variant in a pooled experiment with next-generation sequencing used to count variants in selected populations. From these counts, ratiometric scores are calculated to elucidate the enrichment of each variant in the pool during selection. This requires careful assay design where genetic material encoding the variant and the variant protein itself must co-segregate in any selection step. One way to accomplish this is to measure activity in a cell, where the cell membrane encompasses both the protein variant and the genetic material encoding it and to sort cells with functional variants from those that are impaired.

We elected to develop an assay for PLpro activity in a human cell line to closely mimic the native environment during viral replication. To measure proteolytic activity in cells, we converted an existing luciferase-based reporter[23] into a FRET-based biosensor compatible with fluorescence activated cell sorting (FACS). Our FRET-based biosensor is composed of N-terminal mClover3 donor and C-terminal mRuby3 acceptor fluorophores separated by a linker containing the Nsp2/3 PLpro cleavage motif $T_{P5}L_{P4}K_{P3}G_{P2}G_{P1} \downarrow A_{P-1}P_{P-2}T_{P-3}K_{P-4}V_{P-5}$ (Fig. 1b). We introduced this biosensor into 293 T cells via retroviral transduction to create a stable PLpro reporter cell line. A FRET signal between mClover3 and mRuby3 was readily observable with excitation at 488 nm and emission at 610 nm (Supplementary Fig. 1a).

To validate the biosensor's ability to report on PLpro activity, we installed the PLpro coding sequence into a lentiviral Tet-On expression vector so PLpro expression could be driven by doxycycline (dox) addition. Upon introduction of this construct into our 293 T biosensor cell line, FRET emission was reduced in a dox dependent manner, indicating PLpro dependent biosensor cleavage (Fig. 1b–e). Fluorescence in the donor channel unexpectedly dropped upon biosensor cleavage. The reduction of mClover3 fluorescence indicates that free mClover3 fused to a fragment of the PLpro cleavage motif $(T_{P5}L_{P4}K_{P3}G_{P2}G_{P1})$ is less stable than the intact biosensor. We therefore present data as FRET versus acceptor fluorescence.

Inhibition of PLpro protected the biosensor from cleavage and dose-response curves with two PLpro inhibitors 3k and 5c[18] reported EC50s of 13.8 μM and 10.4 μM (Fig. 1f). We found this approach also worked to create cellular assays for PLpro of other coronaviruses. We designed biosensors that were able to report activities of SARS-CoV, and MERS PLpro, by adjusting the flexible linkers (Supplementary Fig. 1b). Together, these data indicate that our SARS-CoV-2 PLpro biosensor is stable in cells until PLpro expression is induced and that inhibition of PLpro can be faithfully measured, providing a cellular assay that is suitable for DMS.

## Construction and validation of a DMS library

Several methods are reported to generate DMS libraries in high-throughput manner, including nicking mutagenesis[24,25], PCR-based site-directed mutagenesis[26,27], codon-mutagenesis[28,29] and Mutagenesis by Integrated TilEs (MITE)[30]. For residues close to the active site (encompassing residues 62, 69–70, 73–74, 77, 93, 104, 106–119, 151–174, 206–212, 243–253, 260–276, 285–286, 296–304), we ordered dsDNA with gateway adapters from Twist Bioscience and directly cloned these fragments into our lentiviral vector based on FU-tetO-Gateway[31] that had been modified to include a puromycin resistance cassette. To create the remainder of the library, we elected to adapt the MITE method[30] and directly cloned single-strand oligos in tiled cohorts to cover the entire length of PLpro. We additionally included a 16 bp barcode 3' to the PLpro coding region to simplify the sequencing strategy (Supplementary Fig. 2). Once mixed, the full library was characterised by PacBio long-read sequencing to map barcodes to variants. We found 6263 of the 6300 protein variants were present in the library representing 99.4% coverage and each variant was replicated with an average of 13 barcodes, except wildtype which aligned to 6657 barcodes.

## DMS pinpoints structurally and functionally conserved residues

We transduced our DMS library into 20 million FRET-biosensor-containing 239 T reporter cells at a multiplicity of infection of ~0.2 so that most cells that received PLpro only had one copy. Lentiviral virions contain two RNA strands, so to prevent barcode swapping and double transduction of two different variants[32] we mixed our library with an unrelated lentiviral transfer vector containing BFP at a ratio of 1:2 before we made virus. In all, we did seven independent transductions from 4 independent viral productions from a single library plasmid stock. Non-transduced cells were subsequently removed by puromycin treatment, leaving an estimated 2 million unique transductants in each of seven replicates and providing approximately 10-fold representation of the library before selection.

To perform the screen based on biosensor cleavage, we induced PLpro with dox for 24 h prior to sorting cells into FRET positive (inactive PLpro) and negative (active PLpro) gates (Fig. 1g). mRNA was prepared from cells in each population, reverse transcribed and barcodes sequenced by Illumina sequencing. Ratiometric scores for each variant, with modelled errors, were calculated with DiMSum[33] to determine which variants were enriched in the negative FRET gate that contained active PLpro variants.

The data quality was first checked by comparing the distribution of positive and negative controls, which were synonymous variants that encode wildtype protein but have variant DNA codons, and nonsense variants with premature stop codons, respectively (Fig. 2a). We saw good separation of scores, indicating that selection was successful.

We normalized the dataset to place the mean of synonymous wildtype on 1 and nonsense variants on 0, filtered the data to remove variants with low counts and high errors (Supplementary Fig. 3) and calculated a sequence-function heatmap to view the data in its entirety (Fig. 2b, Supplementary Data 1). As expected, no mutations were permitted at the PLpro active site (Cys111, His272, Asp286). Additionally, the four cysteines (Cys189, Cys192, Cys224, Cys226) that tetrahedrally coordinate the zinc in the fingers domain[3] were absolutely required except for Cys224, where another zinc coordinating residue (histidine) was tolerated. This may introduce a non-classical C2-CH zinc finger[34] and thereby maintain structural integrity. All nonsense mutations were inactive except for those that came after Tyr305, which is the end of a major β-sheet (Fig. 2b) in the thumb domain, and residues following Tyr305 may not be required for PLpro structural integrity and hence functionality. Viral genome sequencing over the course of the pandemic has identified naturally occurring PLpro variants[35]. These variants all had scores indicating some level of activity, with all but a handful clustering close to wildtype (Fig. 2c, Supplementary Data 2). Those with lower activity scores and frequency may be sequencing errors in the clinical isolates or may occur in combination with other mutations that restore activity.

To check that our DMS activity scores aligned well with conservation among the PLpro of different coronavirus species, we aligned multiple PLpro sequences from SARS-CoV-2, MERS and other coronaviruses[36] (Supplementary Data 3) and calculated a WebLogo. Although sequence conservation is relatively low between species, some residues are highly conserved. In our DMS dataset these same residues are, almost without exception, intolerant of mutation (Fig. 2b).

As variants with folding defects will have reduced activity, we expect that residues on the surface of PLpro will be more tolerant of mutation than those within the core of the protein. To assess this, we averaged the activity scores of all variants for each position and plotted the resulting score on a ribbon diagram of the PLpro structure, where red indicates tolerant and blue intolerant to mutation (Fig. 3a). In addition to the active and zinc-coordination sites (Fig. 3a) we saw

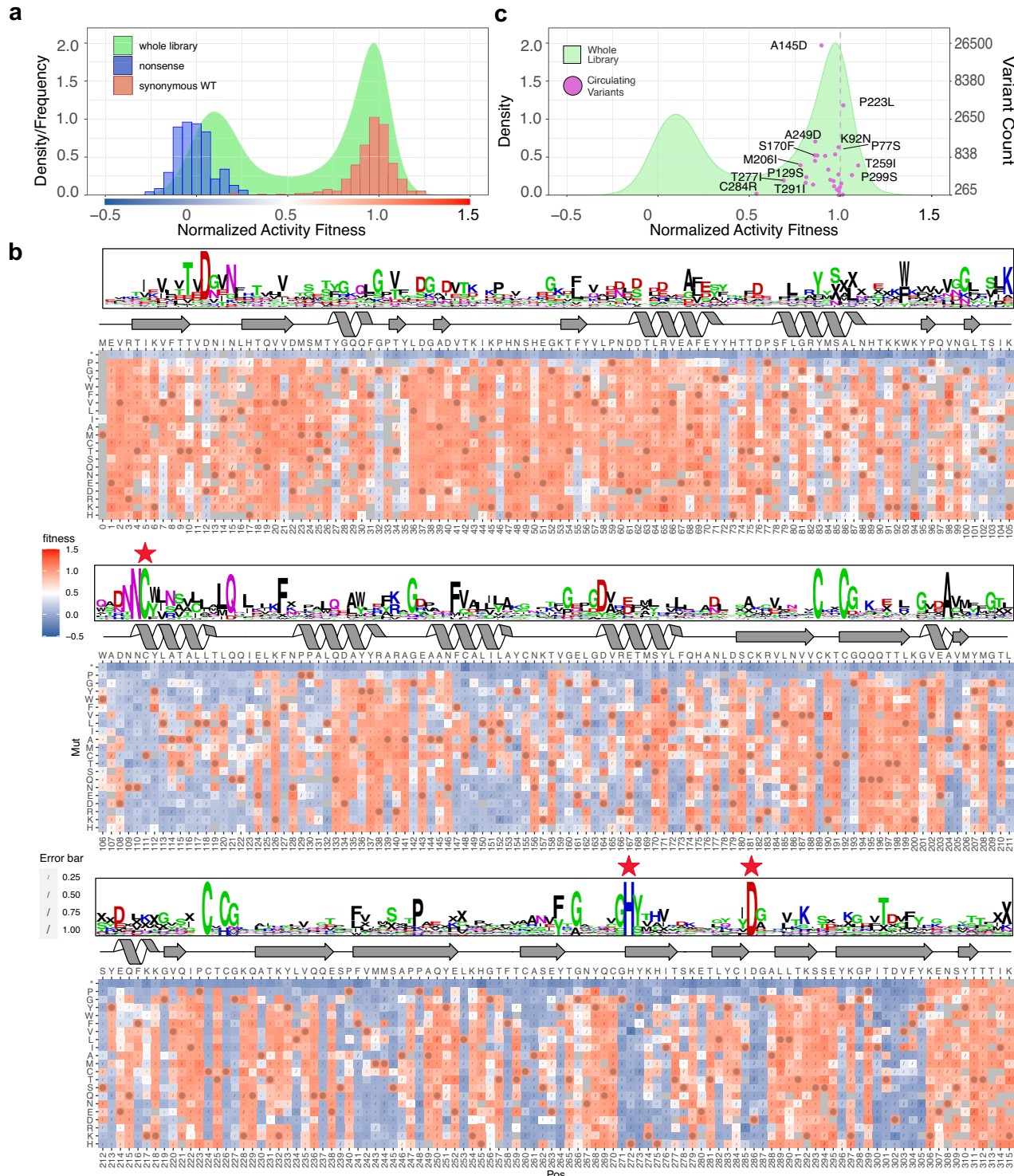

**Fig. 2 | Sequence-Function map of PLpro activity and observed circulating variants. a** The distribution of normalized fitness scores for the whole library (density; green), overlaid with the frequency of scores from synonymous wildtype variants (red; set at 1) and nonsense variants at positions 1-305 (blue; set at 0). The color bar below the x-axis matches the color scheme used in **b**). **b** The normalized sequence-function map of activity data from 7 independent transductions (see SupplementaryData 1 for replicate scores). The bottom x-axis indicates the position (Pos); the top x-axis indicates WT sequence; and the y-axis indicates the mutation (Mut). Fitness is shown in a two-color gradient with red indicating active variants, blue inactive variants, and grey missing variants. Wildtype variants are marked with a circle. Errors (sigma) from DiMSum are indicated with a slash, the size of which scales with the extent of the error. Secondary structure is depicted along the top of the map and based on 7CJM[52]. Arrows denote β-sheet; helices

denote α-helix and lines denote random coil. Three red stars highlight the active triad: C111, H272, D286. On top of the secondary structure diagram, a sequence alignment of 22 PLpro sequences from different coronavirus species (SupplementaryData 3) is shown in weblogo format. Positively charged residues are shown in blue; negatively charged residues shown in red; polar residues are shown in green and magenta; and others in black. "X" was used to fill in gaps in the alignment. The bigger the letter is, the more conserved the residue. **c** The density plot of the whole library in (**a**) was overlaid with the score (x-axis) and variant counts (right y-axis, in log scale) of circulating variants (magenta) identified from COVID-3D[35]. Some commonly observed variants are labeled; refer to Supplementary Data 2 for a full list of variants and scores used in this study. The least active variant C284R is also labeled. Interestingly, the canonical sequence of SARS-CoV PLpro also has an Arg at this position. Source data are provided as a Source Data file.

that residues within the papain-like fold were largely intolerant of mutation (Fig. 3b). In comparison, the N-terminal ubiquitin-like domain is largely tolerant. This agrees with findings from MERS PLpro showing that the ubiquitin-like domain can be truncated with minimal impact on activity[37].

## Identification of expressed variants that affect PLpro function

The DMS dataset measuring PLpro activity has revealed almost all single-site substitutions that either maintain or impact PLpro activity. Impaired variants may carry a mutation that impacts abundance, most likely via folding or stability defects, or function, giving mechanistic insight. To identify abundant but functionally impaired variants, we reformatted the library to install a C-terminal mClover3 fusion (Sup-

count variants, and calculated ratiometric scores to determine the abundance of each variant.

We detected 6130 single site variants in our abundance library, achieving around 97% coverage of all potential 6301 variants and saw segregation among synonymous and nonsense variants (Supplementary Fig. 4b–e, Supplementary Data 4). When activity scores were plotted against abundance scores, we saw good correlation among most data points. We gated inactive variants with WT-like abundance and defined these variants, encompassing 20 residue positions, as structurally intact but functionally impaired (Fig. 3c; Table 1).

We depict these 20 residues on a PLpro structure that includes the LRGG PLpro cleavage motif taken from the PLpro: ubiquitin structure 6XAA[6] (Fig. 3d). Strikingly, mutations that impaired function without

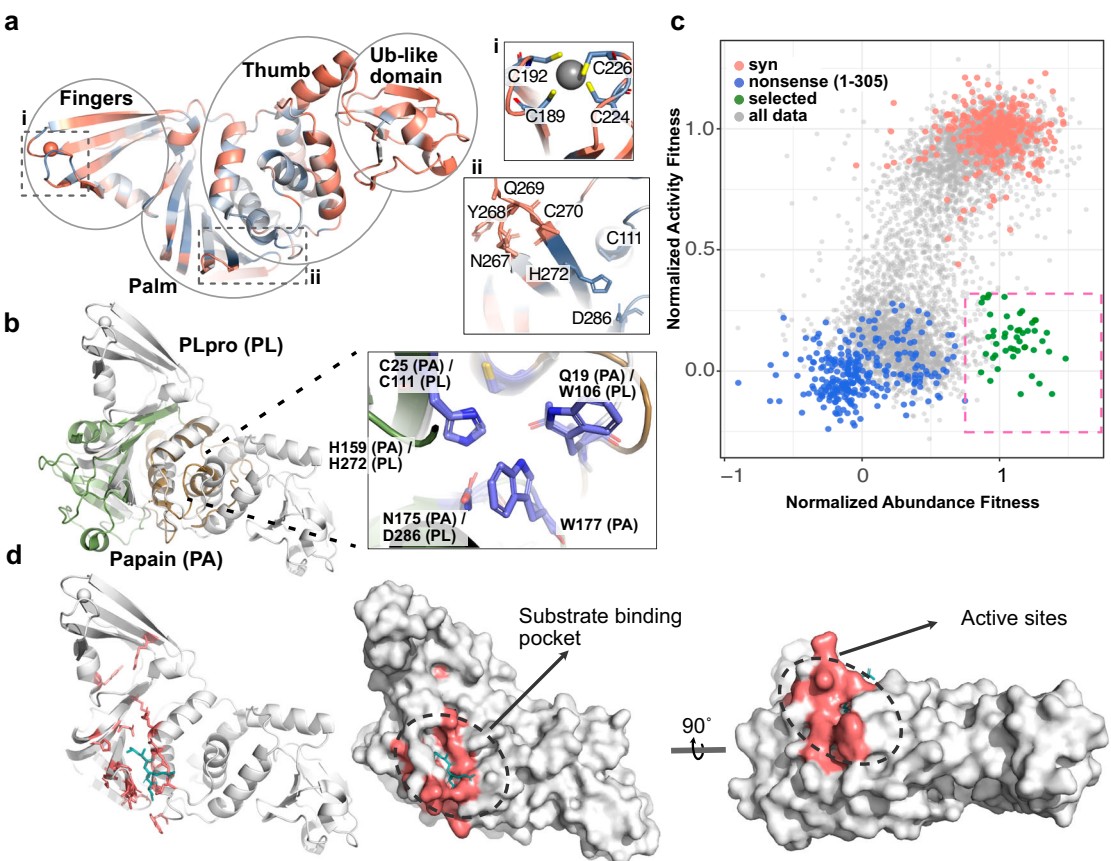

**Fig. 3 | Structural analysis of activity fitness scores before and after filtering for abundance. a** PLpro structure 6XAA[6] in ribbon with the fingers, palm, thumb and ubiquitin-like domains labeled. Two additional regions containing the zinc-coordination site (**i**) and, active site and blocking loop (**ii**) are boxed and expanded to the right. The ribbon and highlighted areas in box i) and ii) are colored according to the average fitness score among all variants at each position. Red positions are tolerant to mutation and blue positions are intolerant. In box (i), C189, C192, C224, C226 are shown in stick and the zinc atom in sphere. In box ii), residues on the blocking loop and active triad (C111, H272, D286) are shown in stick. **b** The papain (9PAP[77]) active site was aligned on the active site of PLpro (6XAA, white) and the regions of papain that showed good structural correlation are shown in green (palm) and brown (thumb). The overlay of the active site is shown on the right. Key PLpro (PL) and Papain (PA) residues are labeled and shown in stick. **c** A scatterplot

of Normalized Activity Fitness Scores against Normalized Abundance Fitness Scores (from two independent transductions, see Supplementary Fig. 4 for Abundance Sequence-Function map) identifies variants with normal abundance, but impaired function. Variants from the whole library are colored in grey, nonsense variants (positions 1-305) in blue, and synonymous wildtype variants (syn) in red. Variants whose abundance score minus its associated error still fall within the box at the lower right of the scatter plot were selected as abundant but impaired and are colored in green. **d** left) The position of abundant but impaired variants are indicated on the PLpro structure in stick and red. The last 4 residues of Ubiquitin are shown in green to highlight the substrate binding region. middle) Surface representation of (d-left). right) PLpro structure in (d-middle) rotated 90° around x to show the active site. Source data are provided as a Source Data file.

plementary Fig. 4), allowing PLpro abundance to be monitored by flow cytometry. Similar methods have been used to measure protein abundance in other systems[38–40]. We expressed this library in parental 293T cells (two independent transductions from one viral stock) and sorted cells from high and low mClover3 gates, recovered mRNA to

impacting abundance were not only found at the active site and substrate binding pockets along the interface of the thumb and palm domains, but also extended all the way up to the fingers domain in a connected pathway. The nature of these mutations and roles in PLpro function are reviewed in the Discussion.

**Table 1 | Classification of functionally important residues**

| Position | Abundant | | | Classification |
| --- | --- | --- | --- | --- |
| | Inactive | Partially active | Active | |
| W106 | C, E, G, K, L, M, R, T, V | A, F, I, N, P, Y | | Oxyanion hole[41]; similar position to Gln19 in papain[49] |
| C111 | A, D, E, G, R, S, T | | | Active site |
| T115 | E | | A, C, N, V | Second shell |
| L162 | D, I, N | H, Q, W | A, C, F, G, K, R, T, Y | Substrate binding[74] |
| G163 | A | | | Interact with substrate C-terminus[52] |
| D164 | A, H, M, T, V, W, Y | G, N | C, E | Substrate binding[52,53] |
| R166 | C, N, T, W, Y | F | | Lines the S4 pocket. Unknown, may stabilise D164[20] |
| R183 | C, E, F, I, L, M, V, Y | | A, G, S | Unknown |
| Y213 | D, E | | F, I, L, M, V, W | Unknown |
| M243 | H, Q, W | | | Unknown |
| P248 | A, C, G | V | | Substrate binding[11,13] |
| Y264 | C, I, K, L, M, Q, W | | F | Substrate binding[44] |
| G266 | K, L, M, N, R | C, H, P, Y | E | BL2 flexibility |
| Q269 | P | D | A, C, E, F, G, H, I, K, L, N, R, S, T, V, W, Y | Substrate binding |
| C270 | P | | A, E, F, G, H, I, L, M, Q, R, S, T, V W, Y | Second shell[75] |
| G271 | A, E, K, R, W, Y | | | Substrate binding[6,52] & blocking loop flexibility, second shell |
| H272 | R, W | | | Active site |
| D286 | E, N | | | Active site |
| G287 | N | | | Second shell |
| D302 | C, N, T | | | Lines the S4 pocket Unknown |

This table reports all positions that have at least one inactive variant that was measured to be abundant. The classifications are performed based on the gates in Supplementary Fig. 20 and full classification of all variants including those that are partially abundant or absent can be found in Supplementary Table 3.

## Predicting drug-escape variants with DMS

The blocking loop of PLpro (spanning residues 267-270) is flexible in its conformation, which likely allows substrates of various size to be accommodated[41]. Our DMS data indicate that almost any mutation in the blocking loop is tolerated for function. Indeed, the MERS PLpro blocking loop differs in sequence and length, indicating that residues at these positions are not highly conserved[42]. This property of PLpro may impact drug-discovery efforts as most reported inhibitors currently engage this flexible region[21,43,44]. We, therefore, decided to employ our DMS screen to systematically investigate whether we could detect drug-escape variants to published PLpro inhibitors 3k, its analogue 5c[41], GRL0617[17], Jun9-84-3[20] and XR8-89[19].

We assessed each compound's activity in our cellular assay (Supplementary Fig. 5; Fig. 1f) to determine an appropriate concentration with which to perform DMS, aiming for at least 80% inhibition. While XR8-89 had good activity in biochemical assays[19], we could only detect activity at 100 μM in cells, indicating this compound may be poorly membrane penetrant. GRL0617 had an EC50 of 37.3 μM, requiring more than 100 μM for an effective inhibition at EC80. At these high concentrations, GRL0617 starts to precipitate and impacts cell viability. Jun9-84-3 has a lower EC50 of 22.8 μM, but like GRL0617, we observed significant impacts on cell viability at high concentrations. Thus, we were only able to assess 3k and 5c in a physiologically relevant cell-based screen. We repeated the DMS screen in the presence of 65 μM 3k (three independent transduction replicates) and 5c (five independent transduction replicates) (EC75−EC85), sorting FRET positive and FRET negative gates for comparison.

The resulting sequence-function maps were assessed for evidence of selection. If drug treatment resulted in 100% inhibition of PLpro activity, we would see similar scores for synonymous wildtype and nonsense variants, as both variants would remain in the FRET positive gate. Instead, we saw that synonymous wildtype variants had higher scores than nonsense variants consistent with incomplete inhibition (Supplementary Fig. 6-7). Nevertheless, some positions showed elevated activity scores, and these were similar among the 3k and 5c

datasets (Supplementary Fig. 8). Because these compounds are almost identical (Supplementary Fig. 5d), we combined the 3k and 5c datasets to gain precision and selected variants for individual testing that would be sensitive to the 3k/5c shared scaffold (Supplementary Fig. 9).

After plotting activity scores 'without compound' against those 'with compound' (Fig. 4a), we selected the following variants for testing in cells: wildtype-like variants L58V and Y268W (which was one of the few residues at position 268 that did not interfere with drug activity), escape variants D164S, D164C, M208A, M208W, and Y268R, as well as M208S, which differed in behavior between 3k and 5c. We transduced 293 T sensor cells with each variant and tested activity before dox treatment (no PLpro), after dox treatment (PLpro alone), and with dox and 65 μM 3k (Supplementary Fig. 10). Almost all inhibitor treatments behaved as expected, with L58V and Y268W remaining sensitive to compound and D164S, D164C, M208A and Y268R showing evidence of drug escape. M208S proved to also escape 3k, as seen in the 5c dataset, indicating that M208S provides an escape route for both compounds. The data from M208W, however, was unusual. The no dox treatment flow-cytometry plots indicated that this variant displayed substantial activity prior to dox-induced PLpro expression (Fig. 4b). Indeed, 3k treatment inhibited dox-induced M208W PLpro back to pre-dox treatment levels (Fig. 4b). This leads us to conclude that M208W was mis-characterized as a drug-escape variant in our screen because of poor dox control. To remove the effect of this artifact from our final list of escape variants, we repeated the screen (two independent transduction replicates) without adding dox to characterize the landscape of leaky expression across our whole library (Supplementary Fig. 11).

We plotted leaky expression fitness scores and observed a non-random distribution of positions in the library with poor dox control suggesting protein intrinsic properties were the basis of this phenomena. Particularly obvious were aromatic residues at positions 170 and 208, which both point into the cleft between the thumb and palm domains, as well residues 313-315 at the C-terminus that are distal from the active site. Since PLpro resides within the multidomain Nsp3

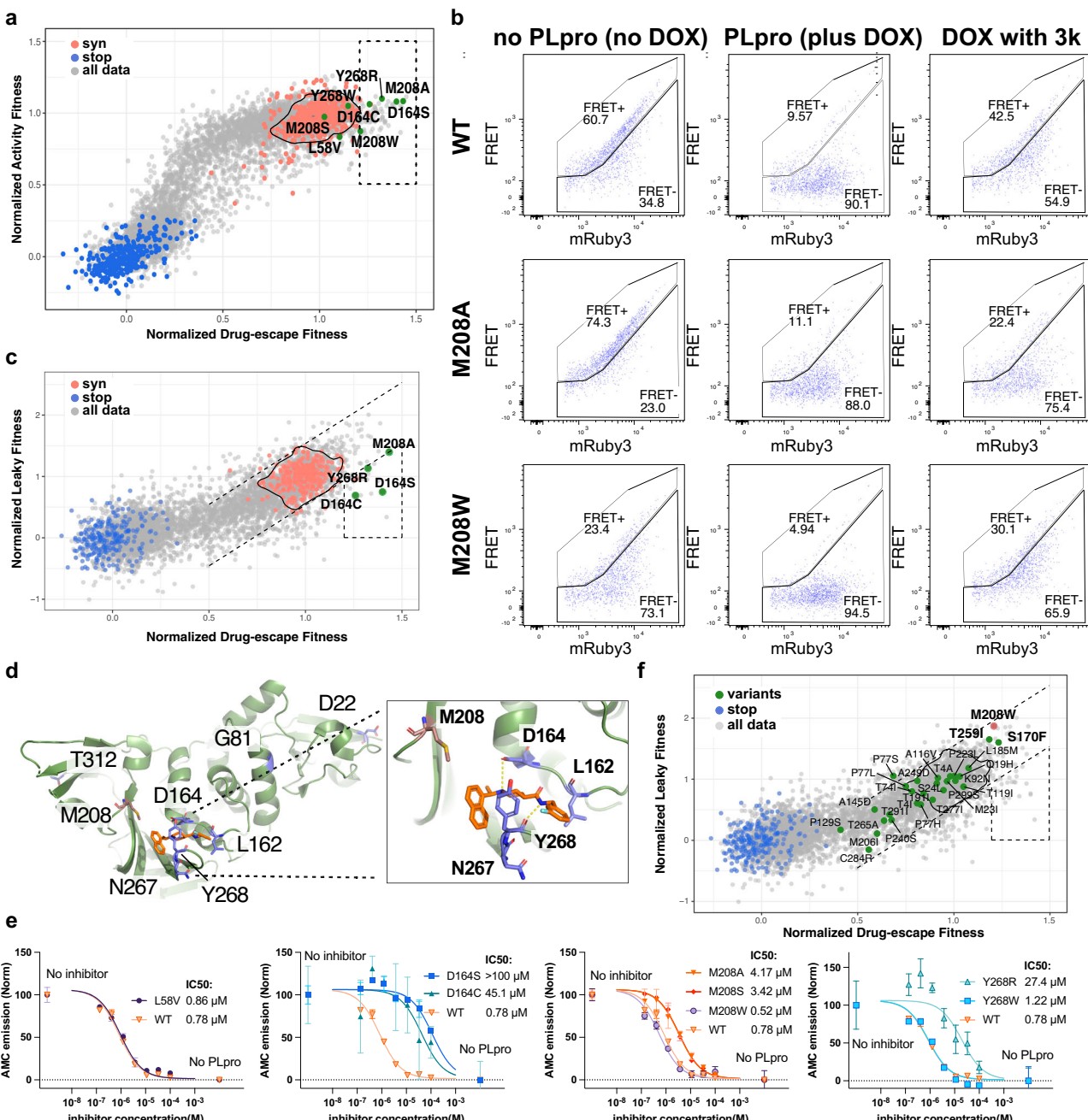

**Fig. 4 | Identification of drug-escape variants to the 3k/5c scaffold. a** Scatterplot of activity against 3k/5c combined drug-escape fitness scores (merged from five 5c and three 3k replicates from 5 independent transductions; See Supplementary Fig. 9 for drug-escape fitness sequence-function maps). Nonsense variants (stop) are shown in blue, synonymous wildtype (syn) in red, and other variants in grey. The density border encompasses approximately 90% of synonymous wildtype datapoints. The dashed box contains variants with drug-escape fitness scores more than the main cluster of synonymous wildtype variants and activity scores similar to synonymous wildtype variants. Variants selected for downstream analysis are highlighted in green. **b** Flow cytometry plots of wildtype, M208A and M208W PLpro with biosensor cells (transduced once, tested twice). 3k was added at a concentration of 65 μM. Other selected variants are shown in Supplementary Fig. 10. Gating strategy is depicted in Supplementary Fig. 10a. **c** Scatterplot of leaky (y-axis) versus 3k/5c combined drug-escape fitness scores (x-axis). See Supplementary Fig. 11 for leaky expression sequence-function map. Dashed lines highlight

the correlation between the two datasets. Variants selected for further analysis are highlighted in green. **d** Positions identified in (**c**) as having drug-escape variants are highlighted in blue stick representation on the PLpro structure (7TZJ)[22]. Positions that contact 3k (orange) and mediate drug-escape are depicted in blue stick. M208 is highlighted in salmon stick. **e** Dose response of 3k with recombinant PLpro using the substrate Z-RLRGG-AMC. Data were scaled with AMC emission without PLpro defining 0% and AMC emission with no inhibitor defining 100% activity. IC50s were calculated from dose response curves with a hillslope of 1. A representative experiment is shown from 2 biological replicates. Error bars from technical triplicates indicate mean ± SD. IC50 values are the average of both biological replicates. **f** Circulating variants (green) identified from COVID-3D[36] were overlaid on the same scatterplot in (**c**). Synonymous wildtype variant datapoints were removed for clarity. The M208W variant is highlighted in red. Source data are provided as a Source Data file.

protein, the C-terminus is not normally exposed. We suspect that the last three residues of the PLpro construct are somewhat destabilizing, and mutation or truncation reduces PLpro turnover in mammalian cells.

The correlation plot of leaky expression versus drug escape fitness scores allowed us to select variants with drug escape scores that were higher than synonymous wildtype variants and leaky expression scores that were in line with or lower than synonymous wildtype variants (Fig. 4c; Supplementary Fig. 12). The mechanism of drug escape for most variants can be explained by contacts to 3k (Fig. 4d). However, some variants were far from the 3k binding site and may indicate noise in our dataset or an allosteric mechanism. The side chain of Met208 appears too far from 3k to make a meaningful contribution to compound binding, yet multiple mutations at this site affect 3k- and 5c-mediated inhibition without impacting PLpro function. This list of Met208 variants (Supplementary Fig. 13-14) includes small side chain mutations, ruling out steric hindrance as a mechanism for drug escape.

## Activity and 3k responsiveness of recombinant PLpro variants

To determine the extent of drug escape, we elected to make recombinant PLpro variants and measure 3k dose-response curves to calculate IC50 scores for each variant. This orthogonal assay confirms that our DMS approach faithfully reports PLpro proteolytic activity and gives an indication of how sensitive our screen is to changes in IC50. It also provides control of PLpro concentration in subsequent assays, eliminating potential differences in expression as a source of drug-escape. We again selected L58V, D164S, D164C, M208A, M208S, M208W, Y268R, Y268W PLpro variants and expressed these variants and wildtype PLpro in *E. coli*. The yields of recombinant protein were similar to wildtype for variants at Leu58, Asp164 and Tyr268. M208A and M208S consistently gave yields around 3-fold lower than wildtype, while M208W yields were approximately 3-fold higher.

The activity of each variant was tested in an assay that measured the cleavage of the commercially available fluorogenic substrate Z-RLRGG-AMC, which contains the PLpro recognition site and emits fluorescence upon cleavage. Unlike our cell-based assay, which measures biosensor cleavage at steady-state 24 h after PLpro induction, this biochemical assay allows the initial rate of substrate cleavage to be measured for each variant and compared to wildtype. While all selected variants had activity scores similar to wildtype PLpro in our DMS assay, we saw impacts on the initial rate at which substrate was processed among variants: L58V and D164S and Y268W behaved similarly to wildtype, D164C and Y268R processed substrate at approximately 30% the rate of wildtype PLpro, and M208A and M208S processed substrate two and half times faster (Supplementary Fig. 15). These results indicate that our DMS assay is powered to identify substantial, rather than mild, defects in activity that we predict will capture more physiologically relevant variants.

We tested each variant for its ability to be inhibited by 3k in the Z-RLRGG-AMC assay. Dose-response analyses were performed and IC50 values extracted from 2 independent experiments (Fig. 4e). The IC50 of 3k against wildtype PLpro was 0.78 μM, while variants selected for wildtype-like activity L58V and Y268W exhibited IC50s of 0.86 μM and 1.22 μM, respectively. Y268R PLpro sensitivity to 3k was reduced, with an IC50 of 27.4 μM. D164S and D164C were the least responsive to 3k, with estimated IC50s of >100 μM and 45.1 μM, respectively. M208A and M208S showed modest loss of activity, with IC50s of 4.17 and 3.42 μM, respectively, while M208W was ~1.4-fold more sensitive to 3k, with an IC50 of 0.56 μM. Together, these data show that we can identify drug-escape variants with as little as 4-fold loss in sensitivity to drug in our DMS analysis.

We compared circulating variants to our drug escape dataset, and none fell in the gate with which we define drug escape variants. However, T259I and S170F variants displayed leaky expression and clustered near M208W (Fig. 4f).

## The impact of Met208 on PLpro structure and function

The position of Met208 in the PLpro structure is interesting in that it sits in the interface between the largely β-sheet-containing fingers domain and the α-helix-containing palm domain, most closely packing against Arg166, a position where we found several inactive variants without a structural explanation. Inspection of PLpro crystal structures with ubiquitin and ISG15[6,45] indicates that Arg166 may make polar contacts with ubiquitin residue Gln49 and ISG15 residue Asn151, but neither appears central enough to the interface to be critical for binding. Furthermore, it is unclear whether Arg166 would play a role in recognition of our cellular FRET-based biosensor. Ma et al., through docking simulations, predict that Arg166 may be mobile and able to create a salt-bridge with Asp164[20]. Asp164 is critical to substrate recognition, making contacts to the backbone of the P4 substrate residue (Leu) in the L(R/K)GG recognition motif that is shared among all PLpro substrates. Thus, perturbation of Arg166 may impair PLpro activity via impacts on Asp164's key role in substrate recognition. Met208, while close to Arg166, sits back from the interface with substrate and does not make intimate contacts with ubiquitin, ISG15 or 3k (Supplementary Fig. 16).

We decided to compare M208A and M208W PLpro Michaelis-Menten kinetics with wildtype PLpro against Z-RLRGG-AMC, Ub-Rhodamine110 and ISG15-Rhodamine110 substrates (Fig. 5a). Surprisingly, M208A significantly boosted enzymatic activity of PLpro against Z-RLRGG-AMC, while decreasing Ub-Rhodamine110 and, to a lesser extent, ISG15-Rhodamine110 cleavage efficiency. M208W performed similarly to wildtype PLpro against each substrate. We were not able to deconvolute $k_{cat}$ from $K_M$ as affinity is predicted to be in the μM range, and we could not achieve these concentrations in our assays. Wishing to further explore the differences between M208A and M208W, we measured thermal stability (Fig. 5b). M208A appeared similarly stable to wildtype PLpro, while M208W strikingly increases the protein melting temperature by over 5 °C, indicating a substantial improvement in thermal stability. Increased stability causing reduced turnover in cells, may provide a mechanism to explain leaky expression in our cellular assay and increased yield of recombinant protein during *E.coli* expression.

To determine whether M208A and M208W altered protein structure, we attempted to crystallize both proteins. Most crystal structures of PLpro contain inhibitor and/or bear the C111S mutation to improve crystal quality. We set up parallel crystallization screens with M208A, M208W and wildtype PLpro, all with intact catalytic sites. M208W readily crystallized (in 14/96 conditions), but we were unable to recover diffraction-grade crystals of M208A or wildtype PLpro. After minimal optimization of screening conditions, we collected a 2.00 Å dataset of M208W with space group and unit cell dimensions ($P2_12_12$; unit cell 58 Å, 85.5 Å, 83 Å, 90°, 90°, 90°) (Supplementary Table 1) that were unique compared to other PLpro structures in the PDB.

The resulting structure was overlaid with the crystal structure of C111S PLpro (7D47) and showed an identical global structure, with an RMSD of 0.495 Å over Cα atoms. The Trp at position 208 had clear electron density (Supplementary Fig. 17) and was accommodated without disturbing neighboring residues. Overlay of the M208W structure with PLpro: Ub, PLpro: ISG15 and PLpro:3k structures indicated that this mutation does not introduce steric clashes with ISG15 or 3k (Supplementary Fig. 16). Val70 of ubiquitin is within 2.5 Å of Trp208, potentially creating a mild steric clash. Nevertheless, M208W cleaved Ub-Rh more efficiently than wildtype PLpro indicating substrate engagement still occurs.

We examined B-factors to determine the extent to which the M208W mutation stabilized local PLpro structure. We collected all PLpro structures in the PDB with a resolution better than 3.5 Å and calculated the average of backbone and Cβ atom B-factors residue-by-residue along the length of the protein. Residues 186–197 and 219–232

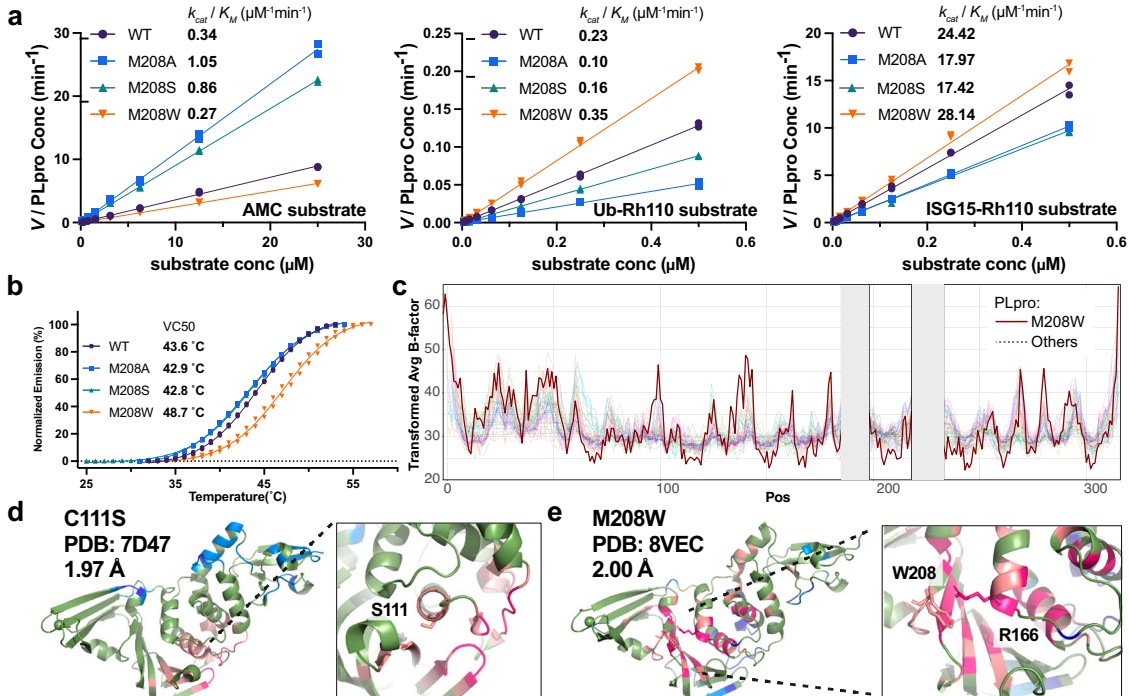

**Fig. 5 | Characterization of M208 variants. a** Michaelis-Menten kinetics of M208 variants using substrates Z-RLRGG-AMC (left), Ubiquitin-Rhodamine110 (middle) and ISG15-Rhodamine110 (right). y-axis is velocity (V) of substrate cleavage divided by PLpro concentration (PLpro Conc) used in the assay. The $k_{cat}/K_M$ is approximated from the slope of the fitted line. Representative experiment from 2 biological replicates. $k_{cat}/K_M$ is the mean from both biological replicates. **b** M208W displays increased thermal stability. The minimum fluorescence value of each curve defines 0% denaturation and the plateau of each curve defines 100% on the y-axis. Representative experiment from 2 biological replicates. The reported midpoints are the mean from both biological replicates. **c** B factors from different 54 PLpro structures (light colors) scaled to the M208W PLpro structure (bold and red line). x-axis shows residue position (Pos) and y-axis is the average of the backbone and Cβ atom B factors after scaling (Transformed Avg B-factor). The grey box masks the zinc-coordinating regions which were excluded from our analysis. **d** C111S PLpro

(7TZJ) is stabilized at residues surrounding the active site. The structure is presented in ribbon and colors indicate regions of stabilization that were less than 2 (pink) and 3 (magenta) standard deviations from the mean of all data and regions of destabilization that were more than 2 (light blue) or 3 (dark blue) standard deviations from the mean. The zoomed view shows the region surrounding C111S. **e** M208W PLpro (8VEC) is stabilized at the interface of the palm and thumb domain. The structure is presented in ribbon and colors indicate regions of stabilization that were less than 2 (pink) and 3 (magenta) standard deviations from the mean of all data and regions of destabilization that were more than 2 (light blue) or 3 (dark blue) standard deviations from the mean. The zoomed view shows the region surrounding M208W and shows a potential role for R166 stacking against the introduced tryptophan at position 208. Both residues are represented in stick. Source data are provided as a Source Data file.

were not considered during the analysis, as these regions have poor density in most reported structures. We plotted B-factor against residue number and calculated scaling factors to normalize the traces using a Broyden–Fletcher–Goldfarb–Shanno (BFGS) iterative algorithm[46] from the "optim" function in R (R studio v4.2.0), which calculates optimal scaling and translation factors to minimize the differences between complex curves. We visually inspected the scaled plots for good overlay (Fig. 5c). If successful, the scaling factors should also similarly normalize the overall Wilson B-factor, and this was the case (Supplementary Table 2).

We looked for local changes in stability as defined by regions where scaled B-factors were 2 (weak) or 3 (strong) standard deviations from the mean of all structures. We first assessed the C111S structure, predicting there would be observable stabilization around the active site. We colored regions of weak or strong stabilization (red tones) and destabilization (blue tones) on the C111S PLpro structure and saw stabilization around the active site as expected (Fig. 5d). In contrast, increased stability in M208W PLpro extended along the whole β-sheet of the fingers domain and propagated into the α-helices of the palm domain that pack against this sheet (Fig. 5e). The area of stabilization encompassed the 3k binding site and may provide a mechanism for improved 3k sensitivity by reducing flexibility without perturbing structure. While we were unable to determine the structure of M208A, we predict that Ala at this position would be destabilizing compared to

methionine. We did not see this manifest in the thermal shift assay measuring the temperature at which PLpro denatures, but this prediction is consistent with lowered yield of recombinant protein when this variant was expressed in *E. coli*.

## Discussion

In this study, we comprehensively examined the mutational effects of almost every PLpro residue on the proteolytic activity and abundance of SARS-CoV-2 PLpro. Our data are validated by multiple observations. We see that mutation of the active site ablates PLpro activity, as does introduction of premature stop codons anywhere in the sequence except the last 10 residues. We also confirmed the role of the zinc-coordinating cysteines in the fingers domain as critical for protein integrity[3], as variants at the four zinc-coordinating cysteines (with the exception of C224H) destroy proteolytic activity and reduce PLpro expression. By comparing activity and abundance datasets, we identified 20 positions where the function of PLpro is impaired without impacting structure. The nature of these positions is discussed below.

PLpro is a member of papain-like thiol proteases superfamily. Papain itself is the model protein on which the catalytic mechanism of proteolytic cleavage can be understood for this family. Papain is a globular protein with L- and R-domains. Papain shares very little sequence homology with PLpro, but when the structures are overlaid

based on the arrangement of residues in the active site, parts of the L- and R- domains, and particularly the interface between them, overlay well on the thumb and palm domains of PLpro (Fig. 3b). Like PLpro, the active site of papain is split between the L- and R-domains, with Cys25 on the L-domain and His159 found on the R-domain, which correspond to C111 and His272 in PLpro, respectively. The catalytic dyad formed by Cys25 and His159 is the minimal requirement for papain-mediated catalysis and results in a Cys⁻-His⁺ ion pair that acts as a nucleophile during protein cleavage. Three additional residues play prominent roles in catalytic activity. Asn175, often thought of as part of the catalytic triad, is not strictly necessary for catalysis, but rather plays a role in positioning His159 and modulating the ionization state of the catalytic dyad by influencing the pKa of His159[47]. Trp177 also modulates the ionization state and plays a part in protecting the Asn175-His159 interaction from solvent[48]. Glu19 stabilizes a transition state intermediate of the substrate during cleavage and forms part of the oxyanion hole that releases charge buildup during substrate cleavage[49]. In the PLpro active site, Asp286 plays the role of Asn175 in papain, Trp106 occupies the same relative position as the oxyanion-hole residue Gln19 in the papain active site, and there is no equivalent for papain residue Trp177.

Our DMS data confirm the absolute requirement for Cys111 and His272 in catalytic activity. Given that Asp286 is equivalent to Asn175 in papain, we were surprised that not even Asn is tolerated at this position in PLpro. Our alignments indicate that Asp is used exclusively in viral PLpro sequences. Asp instead of Asn may be preferred because of the lack of a PLpro residue analogous to Trp177, which shields the papain Asn175-His159 interaction from solvent.

PLpro residue Trp106 occupies the same position as oxyanion hole residue Gln19 and mutation in SARS-CoV PLpro impairs PLpro function[50]. While this Trp can be mutated to almost any other residue without affecting PLpro abundance, PLpro activity is only maintained with the substitution of polar residues Gln and Asn, other aromatic residues Tyr and Phe, and Ala. Although Trp is preferred, the nature of the mutations tolerated at this position is consistent with its ascribed role in stabilizing an oxyanion intermediate as suggested by Baez-Santos et al. in their analysis of SARS-CoV PLpro[41]. MERS PLpro has a Leu at this position[42], which is unable to behave as a hydrogen-bond donor and our results indicate a similar mutation would impair SARS-CoV-2 PLpro. One study mutated MERS PLpro Leu106 to Trp and saw an increase in MERS PLpro activity, concluding that the properties of the oxyanion hole of MERS PLpro limit catalytic efficiency[51].

Our assays identify the residues required for cleavage of the Nsp2/3 site defined as $L_{P4}K_{P3}G_{P2}G_{P1} \downarrow A_{P1'}$. We identify features of substrate binding pockets and the sequence requirements of the blocking loop, which lines the narrow channel that feeds the P2 and P1 glycines of the substrate to the active site.

The blocking loop is flanked by Gly266 and Gly271. Both glycines are highly conserved among coronaviruses and Gly271 is completely intolerant of mutation in our system. Addition of a sidechain to Gly271 would impact blocking loop conformation and accommodation of the P2 Gly of substrate. Being adjacent to the catalytic His272 means perturbation of Gly271 would impact the active site. We thus see Gly271 is both functionally and structurally important. Gly266 variants are partially functionally impaired, but abundance is not reduced, indicating a likely role for flexibility of the blocking loop for substrate binding. The sequence between Gly266 and Gly271 is tolerant to almost any mutation, indicating that movement of the blocking loop is likely required, but the sidechains of the blocking loop are dispensable for substrate binding and PLpro activity.

The S4 pocket that binds the essential P4 Leu is larger than what is needed to accommodate Leu and has been exploited for drug-discovery purposes instead of the shallow active site. The S4 pocket is lined by Asp164, Arg166, Met208, Pro247, Pro248, Tyr264, Tyr273, Thr301 and Asp302; but only Asp164, Pro248, Tyr264 and Tyr273 and

Thr301 are in close proximity to the P4 Leu. The blocking loop flips up and forms a roof over the S4 pocket. Among the residues near the P4 Leu, we find functional roles for all except Thr301 and Tyr273, which both play structural roles. We also see a functional role for Arg166 and Asp302, despite their distance from the site of P4 engagement, that cannot be explained by a structural impairment in our datasets. The importance of these residues towards substrate binding have been described elsewhere[6,11,13,17,52–54].

The substrate's P2 Glycine is situated within the narrow passage leading to the active site. This passage is bordered on one side by Glycine 271 (discussed above), and on the other side by Glycine 163. Additionally, Leucine 162 is positioned directly over P2. We found that Gly163 was structurally intolerant to all mutations except Ala, and G163A was functionally impaired, highlighting that even a single methyl group is sufficient to block the substrate channel. The chemistry of Leu162 was also seen to be functionally important with short polar side chains being disfavored.

We also identified that Arg183, Val184, Tyr213, and Met243 are important for PLpro function, and with Arg166 and Asp302, they form a connected pathway from the fingers domain all the way to the S4 pocket. Arg166 has been proposed to move upon substrate binding and stabilize Asp164, a key residue for substrate binding[20]. It is possible that these residues, with Met208 (see below), play an allosteric role in modulating PLpro engagement with substrate.

PLpro is a validated drug target for COVID-19 treatment[6]. We took advantage of our robust mammalian cell DMS platform to explore how PLpro might evade inhibition from the 3k/5c scaffold, which were the only reported compounds we found that could be used sensibly in our cellular assay. In doing so, we provide the framework for assessing other more potent lead compounds as they emerge. SARS-CoV-2 PLpro has an active patent landscape with 45 patents[43] filed since the start of the pandemic. We attempted to measure resistance to GRL0617, Jun9-84-3 and XR8-89 but could not achieve high enough concentrations in our cellular assays without encountering toxicity and precipitation issues. We therefore limit the findings reported here to the two related compounds 3k and 5c.

The PDB contains many entries of PLpro in complex with various inhibitors, and almost all contact Tyr268 on the blocking loop. Our DMS datasets indicate that this residue is dispensable for PLpro proteolytic activity, although our experiments with recombinant PLpro measuring initial rates of substrate cleavage indicate that Y268R activity is substantially lower than wildtype. We were able to correlate our DMS Activity scores to viral fitness through assessment of circulating variants: C284R has been observed in patient samples, and raw DMS activity score indicates an approximately 15-fold reduction in activity compared to wildtype. Although rare in SARS-CoV-2 circulating variants, SARS-CoV PLpro contains an Arg at this position, providing support for this being a genuine circulating variant and indicating that even poor PLpro activity can result in functional virus.

Our initial drug escape data highlighted that Met208, Tyr268 and Asp164 are important to drug binding. We validated several variants at these sites and were able to confirm all but one indeed resulted in drug escape. M208W was the exception, and we identified that poor dox control contributed to its misidentification. In response, we created a map of variants that had higher than expected expression in the absence of dox and surmise that these variants will either have increased catalytic activity or increased protein stability.

We were then able to exclude variants with high leaky expression to isolate true drug escape variants. Because several filters have been applied to generate our final datasets, and some variants are missing from the analysis, we do not claim that this is a complete list of drug escape variants. Nevertheless, of the variants we can confidently define, most are found with mutations at Tyr268 and Asp164, both of which directly contact the 3k scaffold: Tyr268 via a T-shaped π stack against the 3k naphthalene and Asp164 through a hydrogen bond to

the piperidine nitrogen[21]. Aromatic residues at Tyr268 were found to retain sensitivity to drugs, while other mutations provide escape. At Asp164, only Cys, Ser, Asn and Gln variants are functionally tolerated, and of those, D164C and D164S cause drug escape.

The DMS also revealed Met208 as being important to 3k and 5c activity. The reason for this was unclear, since Met208 does not directly contact either inhibitor, with the terminal Cε atom of Met208 being 4.4 Å from the nearest carbon atom of the naphthyl ring of 3k (7TZJ)[21]. By inspecting the crystal structure of PLpro with ubiquitin (6XAA) and full length ISG15 (7RBS)[6,19], we conclude that Met208 also does not play an essential role in these interactions. Our functional DMS data additionally shows Met208 to be tolerant of almost any other substitution. Other coronaviruses have Phe and Ala at this position, which further demonstrates that Met208 is neither conserved nor required for proteolytic, deubiquitination or deISGylation activities. We confirmed M208A and M208S reduced drug sensitivity in both cellular and biochemical assays with recombinant protein, but we observed that M208W was sensitive to drug, which excludes a role for steric hindrance as a mechanism for drug escape. Instead, we propose that thermal motion around the S4 pocket is important. M208A and M208S both increased proteolytic activity of PLpro while modestly decreasing the rate of ubiquitin or ISG15 cleavage. M208W behaved similarly to wildtype PLpro in assays measuring catalysis, but it promoted thermal stability as measured by an increase of 5 °C of the PLpro melting temperature, and it readily crystallized, which through B-factor analysis showed stabilization of an area encompassing large parts of the palm and thumb domains. Together, mutations at Met208 suggest that there is role for thermal motion in substrate preference and may explain why mutations distant from the active site, and extending to the fingers domain, impact PLpro activity.

Similar to work performed by Bolon and colleagues[55], who determined the fitness landscape of Mpro, we applied DMS to decipher the mutational landscape of PLpro and deepen our understanding of the biology of this crucial viral protease. Our results further provide knowledge of the potential escape routes open to the virus to evade developing therapeutics. Plasticity in the S4 binding pocket and blocking loop presents a hurdle that should be considered during the drug development process. This therefore benefits current and future efforts in developing PLpro inhibitors and opens the door for other research into the catalytic mechanism of PLpro.

## Methods

### Vectors
The FRET biosensor and wildtype SARS-CoV-2 PLpro coding sequences were ordered as G-blocks (IDT) and subsequentially installed into the Gateway donor vector pDONR221 (Thermo Fisher Scientific, #12536017) with BP clonase (Invitrogen, #11789020). The FRET biosensor was shuttled into pMX-Gateway-IRES-Hygro (gift from Andrew Brooks, University of Queensland) with LR clonase (Invitrogen, #11791020). The PLpro recipient vector, FU-tetO-Gateway-rtTA-2A-Puro (Supplementary Fig. 2c), was constructed from FU-tetO-Gateway, a gift from John Gearhart (Addgene plasmid #43914; http://n2t.net/addgene:43914; RRID:Addgene_43914), pMSCV-puro[56] and pFTRE3G[57]. Briefly, the PGK driven Puromycin resistance cassette from pMSCV-puro was PCR amplified with oligos containing XmaI sites. The PCR product was cut with XmaI (NEB, #R0180S) and ligated into AgeI (NEB, #R3552S) cut FU-tetO-Gateway to generate FU-tetO-Gateway-Puro. The rtTA-2A-Puro was then installed by cutting FU-tetO-Gateway-Puro and pFTRE3G with AgeI and BsiWI (NEB, #R3553S). The final vector, FU-tetO-Gateway-rtTA-2A-Puro, was created by ligation of the vector fragment from FU-tetO-Gateway-Puro and the insert fragment of pFTRE3G. The pOPIN-B vector was used for recombinant PLpro expression and was a gift from Ray Owens (Addgene plasmid #41142). All oligos used in this study were ordered from IDT. All individual PLpro mutations were introduced into pDONR221-PLpro using NEB Q5

Site-Direct Mutagenesis Kit (NEB, #E0554S) following manufacturer's instructions and shuttled into FU-tetO-Gateway-rtTA-2A-Puro using LR clonase. We also constructed pFGH1-UTG-mTagBFP2 as empty vectors by inserting PCR-amplified mTagBFP2 from pLKO.1−TRC (a gift from Timothy Ryan; Addgene plasmid #191566) into digested backbone from pFGH1-UTG, a gift from Marco Herold (Addgene plasmid #70183) via HiFi assembly (NEB, #E2621X). This empty vector is used to co-package lentivirus for PLpro DMS and prevent barcode swapping.

### Cell line
Verified HEK293T cells were sourced from Cellbank Australia, (#12022001) and maintained in DMEM (GIBCO, #10313039) supplemented with glutamine and 10% FBS (GIBCO, #2526728RP).

### Retroviral and lentiviral production and transduction
HEK293T cells were transfected with transfer vector and retroviral packaging vectors MMLV-gag-pol and VSVg or lentiviral packaging vectors RSV-REV, pMdi and VSVg (gifts from Marco Herold, WEHI) with Calcium Phosphate. Supernatant was harvested 2 days later and filtered through a 0.45 μm filter. Polybrene (Sigma Aldrich, #TR-1003-G) was added to the viral supernatant to a final concentration of 4 μg/ml. Virus was diluted in complete DMEM + polybrene to achieve the desired MOI and added to HEK293T cells at 50% confluency. Cells were spun at 37 °C for 45 min at 1000 g to encourage transduction before overnight incubation at 37 °C. Media was replaced the following day and cells assessed or placed on selection two days post transduction.

### Generation of HEK293T biosensor cells
pMX-mClover3-TLKGGAPTKV-mRuby3-IRES-Hygro was packaged in HEK293T cells with MMLV-gag-pol and VSV-g. HEK293T cells were transduced at a multiplicity of infection (MOI) of approximately 0.1 to avoid multiple integrations. Two days after transduction, the cell line was subjected to 180 ng/ul Hygromycin (Merck, US1400052) treatment for 2 days to remove non-transfected cells and sorted twice based on medium FRET signal. The resulting cell line was aliquoted in cryoprotective medium containing 90% FBS + 10% DMSO (Sigma-Aldrich, #472301-100 ML) and subsequently stored in liquid nitrogen. A fresh batch of cells was thawed for every new experiment. The same procedure was repeated for biosensors with SARS-CoV and MERS PLpro cleavable linkers.

### Introduction of PLpro into HEK293T biosensor cells
FU-tetO-PLpro-rtTA-2A-Puro was packaged in HEK293T cells with RSV-REV, pMdi and VSVg. Retrovirus was transduced into HEK293T biosensor cells at a multiplicity of infection (MOI) of approximately 0.1 to avoid multiple integrations. Two days after transduction, the cell line was subjected to 2 ng/μl Puromycin (Thermo, #A1113803) treatment for 2 days to remove non-transfected cells. The resulting cell line was aliquoted in cryoprotective medium containing 90% FBS + 10% DMSO and subsequentially stored in liquid nitrogen. A fresh batch of cells was thawed for every new experiment. The same procedure was repeated for SARS-CoV and MERS PLpro.

### Inhibitor dose response assay
Wells of a 96-well flat-bottom plate were seeded with $2.5 \times 10^4$ cells in 150 μl media. Compound was serially diluted in DMSO in a 7 point, 3-fold dilution starting at 10 mM. 1.5 μl each dilution was added to the appropriate wells and 300 ng/ml dox (Sigma Aldrich, #D5207) added to induce PLpro expression. Each compound was tested in triplicate. In every drug screen, 5c was used as benchmark. After overnight incubation at 37 °C, 10% CO₂, cells were detached and analyzed by flow cytometry (WEHI FACS facility) to determine the FRET+ percentage of cells at each inhibitor concentration.

To fit dose-response curves non-linear regression was used to fit the data from 5c using Eq. 1 in Prism 9. The Hill slope was set to 1 and

constants determined for the Top and Bottom of the curve:

$$FRET\% = Bottom + \frac{(Top + Bottom)}{\left(1 + \left(\frac{EC50}{[\text{inhibitor concentration}]}\right)^{Hill\ slope}\right)} \quad (1)$$

The data from all compounds in each experiment was then normalized by calculating the scaling factor ($k$) and translating factor ($b$) and correcting FRET+ % to PLpro activity using Eqs. 2 and 3.

$$k = \frac{100}{(top - bottom)}; b = -1*k*bottom \quad (2)$$

$$PLpro\ Activity = k*[FRET\%] + b \quad (3)$$

For other compounds non-linear regression performed on normalized data to determine EC50s with the top set at 100 and bottom as 0. The Hill slope was varied from 1 as required.

## PLpro library construction

For residues close to the active site (encompassing residues 62, 69-70, 73-74, 77, 93, 104, 106-119, 151-174, 206-212, 243-253, 260-276, 285-286, 296-304) we ordered dsDNA with gateway adapters from Twist and directly cloned these fragments into our lentiviral vector FUV-tetO-Gateway-rtTA-2A-Puro vector. For the remaining residues, we adapted MITE mutagenesis from ref. 31 (Supplementary Fig. 2a). The library was broken up into 7 cohorts of approximately 40 residues each. Three fragments were cloned in a single HiFi Assembly reaction (NEB, #E2621X). 1) The vector sequence was PCR amplified from pDONR221-PLpro and spanned a region 3' to the barcode (installed 3' to the PLpro stop codon) to 5' to the region targeted for mutagenesis. 2) A PCR product was created that spanned 3' to the region targeted for mutagenesis to the end of PLpro and contained 16 bp barcode. 3) The region targeted for mutagenesis as ordered as an IDT oPool and contained NNK variant codons. All fragments had overlaps of ~25 bp to facilitate HiFi Assembly. The Assembly reactions were transformed into DH-10b electro-competent cell (NEB, #C3020K) and split into 2% and 98% before spreading on Agar plates containing 50 ng/µl Kanamycin (Gibco, #11815032). Colonies on the 2% plate were counted manually and used to estimate the total number of transformants. Collected transformants were determined by the cohort size aiming for approximately 10 colonies per DNA variant. The cohorts were moved to FU-tetO-Gateway-rtTA-2A-Puro by Gateway LR cloning. All cohorts and the Twist library were pooled at equal molarity.

## PLpro abundance library construction

We shuttled the PLpro library back to pDONR221 with BP clonase to create a complete library in the pDONR221 backbone. We designed a PLpro-mClover3 fusion separated by the linker PVGGSGGGGSGGGG and ordered this as an IDT G-block and cloned it into pDONR221 with BP clonase (Invitrogen, #11789020) to be the basis of our new abundance library.

To construct this library, we cut our existing PLpro library (in the pDONR221 backbone), with PpuMI (NEB, #R0506S) (at G298) and EcoRV (NEB, #R0195S) (after PLpro barcode) and recovered the PLpro library fragment. Meanwhile, we PCR amplified the pDONR221-PLpro-linker-mClover3 to generate a product containing the vector sequence and a second product spanning 3' region of PLpro appended with a 16 bp barcode. The two PCR products were mixed with the library fragment and assembled with HiFi assembly (NEB, #E2621X). This strategy maintains variants from the first position to L290. We then created new variants between L290 and K315 with an additional round of MITE mutagenesis as described in the previous section, but now using pDONR221-PLpro-linker-mClover3 as the source of vector.

## PacBio characterization of DMS Libraries

The PLpro library was cut with NotI (NEB, #R0189S) and NdeI (NEB, #R0111S) to release a 3908 bp fragment containing PLpro and the barcode. The fragment was gel purified and isolated with a ZymoClean Gel extraction kit (Integrated Science, #D4008). The sample was lyophilized and sent to AGRF (University of Queensland) for further sample preparation and sequencing on the Revio system (PacBio).

We obtained approximately 4 million high-fidelity consensus reads. After trimming to the region of interest and correcting orientation of reverse reads with cutadapt version 3.4[58], reads were further filtered with cutadapt to remove reads with greater than 100 mutations. Then barcode error correction was performed with UMI tools version 1.1.4[59] with an error-correct-threshold setting of 2 and only sequences with more than one read were kept. To generate the barcode look up table, the barcode corrected fastq file was loaded into R studio v4.2.0 with readfastq from the "ShortRead" pacakge[60]. The reads were separated by barcode and global pairwise alignment[61] was performed. If 50% or more of the reads for any given barcode had an insertion or deletion at the same position, that barcode was discarded. For a variant to be called for a particular barcode, greater than 90% of the reads had to contain the same mutation. Barcodes attached to reads without mutations were classified as wildtype. Barcodes with reads that didn't meet these criteria were discarded. Finally, a barcode lookup table for the library was constructed from the reads that passed the criteria. We identified 102,363 unique barcodes for the PLpro library. As library construction involved two cloning methods, we also plotted the read distribution per variant for each method to confirm that there is no systematic bias that would affect our results (Supplementary Fig. 18).

## Nanopore characterization of the PLpro abundance Library

The library was cut with MluI (NEB, #R3198) and EcoRV (NEB, #R0195S) to release a 2416 bp fragment containing PLpro-mClover3, which was then prepared by the WEHI Genomic Hub for Nanopore (Oxford Nanopore Technologies) sequencing with SQK-LSK114 kit and R10.4.1 flow cell. Around 26 million reads were obtained and basecalled with dorado version 0.5.2 (model: dna_r10.4.1_e8.2_400bps_sup@v4.3.0). Cutadapt version 3.4[58] was used to trim the reads to only include PLpro, mClover3 and a short linker to the barcode. Concurrently we performed Illumina sequencing of the PLpro abundance library and built a barcode whitelist utilizing UMI tools version 1.1.4[59] with an error-correct-threshold setting of 2 and minimum reads amount of 2.

We summarized the occurrence of unique barcodes in Nanopore reads. Only barcodes with more than 2 reads were kept. Each barcode was compared to those in the whitelist by calculating the Levenshtein distance. If the Nanopore barcode differed by no more than 2 edit distance from only one barcode in the whitelist, it was corrected to match the whitelist barcode. Otherwise, the reads were discarded.

Once all barcodes were corrected, reads were partitioned based on their barcodes using seqkit v2.61[62] and proceeded to a consensus sequence generation pipeline adapted from Subas Satish et al. [63]. Reads were first aligned with minimap2 version 2.17-r941[64]. The consensus sequence was generated using medaka version 1.113 after being refined with racon version 1.5.0[65]. We identified 146,972 unique barcodes. Among them, 105,942 barcodes aligned to a single site substituted PLpro variant, leading to an average of 16.8 barcodes per variant. The Nanopore read distribution among variants is shown in Supplementary Fig. 19.

## DMS library selection

To ensure cells received a single copy of PLpro we mixed the libraries with an unrelated lentiviral vector pFGH1-UTG-mTagBFP2 at a 1:2 ratio. This strategy has previously been reported and minimizes barcode swapping during viral production and transduction[32]. Virus containing the PLpro library was then transduced into HEK293T biosensor cells, or

in the case of the mClover3-containing abundance library, parental HEK293T cells, ensuring a 10-fold replication of barcodes at an MOI of approximately 0.2. Two days after transduction, transduced cells were enriched in 2 ng/μl Puromycin (Thermo, #A1113803) for two days. Live cells were then harvested and seeded for respective selections.

For the activity screen, we treated cells with 300 ng/ml dox for 24 h and sorted FRET positive from FRET negative cells for mRNA extraction (Fig. 1g). For the abundance screen we similarly treated with dox but sorted on mClover3 high and mClover3 low cells (Supplementary Fig. 4). To select for drug-escape variants, cells were similarly treated with dox and 3k and 5c were added to a concentration of 65 μM corresponding to an effective concentration of ~ 80%. 24 h later, cells were sorted from the FRET positive and FRET negative gates for mRNA extraction. To measure the efficiency of dox control (leaky expression), cells were not treated with dox prior to sorting from the FRET positive and FRET negative gates but were instead treated with dox for 4 h after sorting to induce mRNA for extraction.

### mRNA extraction and next-generation sequencing
RNA was extracted with RNeasy Mini Kit (Qiagen, #74106) following the manufacturer's instructions. The RNA was reverse transcribed with a primer that annealed 3' to the barcode and contained an overhang adapter for index primer annealing. After reverse transcription the cDNA was amplified by a second overhang primer annealing 5' to the barcode and 3' to the PLpro stop codon and a reverse overhang primer. The resulting PCR product was indexed in triplicate using the flanking overhangs to allow multiplexed sequencing. The indexed samples were sent for NextSeq 2000 (illumina), single-end sequencing at the WEHI Genomics Hub. The average reads per barcode for each sample was typically more than 30.

### Illumina data processing and variant scoring
The data were demultiplexed and trimmed with cutadapt version 3.4[58]. The final data consisted of fastq files containing barcodes representing the frequency of each variant's mRNA in the selected population. Variant fitness scores and associated errors from various selection conditions (see below) were calculated with DiMSum[33] using Illumina-derived barcode fastq files in combination with long read-derived barcode lookup tables to deconvolute barcodes into variants. The quality filter was set at 15 and no minimum read number was set. Technical replicates were used to combine samples with triplicate indexes after checking for concordance. Experimental replicates were defined as selections from independent transductions. DiMSum automatically normalizes scores such that wildtype is 0. Instead, we wanted to normalize data to center the scores from barcodes linked to wildtype PLpro to a score of 1 ($F_{WT}$) and the population of nonsense variants to a score of 0 ($F_{avg-nonsense(res1-304)}$) for easy visualization of variants and comparison of datasets. To accomplish this, raw DiMSum fitness scores $F_{raw}$ were normalized with a scaling factor ($k$) and translation constant ($b$) according to Eqs. 4 and 5.

$$k = \frac{1}{\left(F_{WT} - F_{avg-nonsense(res\,1-304)}\right)}; b = -1*k*F_{avg-nonsense(res\,1-304)} \quad (4)$$

$$F_{scaled} = k*F_{raw} + b\,5 \quad (5)$$

Errors were scaled with $k$ according to Eq. 6.

$$Sigma_{scaled} = k*Sigma_{raw} \quad (6)$$

### Activity scoring
Samples from the FRET positive gate (inactive PLpro) were treated as time-point 0 in DiMSum, while those from the FRET negative gate

(active PLpro) were treated as timepoint 1. After normalization, data was filtered to remove variants with no more than 10 reads at time-point 0 and sigma no less than 0.2 (Supplementary Fig. 3). Positive fitness scores indicate activity and negative fitness scores indicate impairment.

### Abundance scoring
Samples from the mClover3 low gate (misfolded PLpro) were treated as time-point 0, while those from the mClover3 high gate (folded PLpro) were treated as timepoint 1. After normalization, data was filtered to remove variants with no more than 10 reads at time-point 0 and sigma no less than 0.45 (Supplementary Fig. 4). Positive fitness scores indicate expression and negative fitness scores indicate impaired expression.

### 3k and 5c drug-escape scoring
Samples from the FRET positive gate (sensitive and functionally impaired PLpro) were treated as time-point 0, while those from the FRET negative gate (insensitive PLpro) were treated as timepoint 1. After normalization, data was filtered to remove variants with no more than 5.5 (3k) and 9 (5c) average reads and errors no less than 0.25. Fitness scores greater than 1 indicate reduced sensitivity to drug, while scores below 1 indicate inhibition and/or impaired PLpro activity.

### Characterizing each variant's leaky expression profile
Samples from the FRET positive gate (dox-controlled or inactive PLpro) were treated as time-point 0 in DiMSum, while those from the FRET negative gate (poor dox control) were treated as timepoint 1. After normalization, data was filtered to remove variants with no more than 18 reads at time-point 0 and sigma no less than 0.5 (Supplementary Fig. 11). Positive fitness scores indicate leaky expression and negative fitness scores indicate good dox control or impaired PLpro.

### Calculation of sequence-function maps
Sequence-Function maps with blue-white-red gradients representing scores and lines representing errors were prepared in R studio v4.2.0 with ggplot2. For activity and abundance scores the white point was set to 0.5 (midway between synonymous wildtype and nonsense variant scores). For drug-escape and leaky expression scores the white point was set at 1 (centered on wildtype scores).

### Identification of functionally important variants
We plotted normalized abundance scores against normalized activity scores to find functionally important variants that remained abundant but had impaired activity. Based on the distribution of wildtype and nonsense variants with synonymous mutations, we classified variants with abundance scores exceeding 0.75 and activity scores below 0.32 as functionally important. Given the proximity of many variants to the threshold and variations in error magnitudes, we additionally mandated that variants adhere to these criteria even after adjusting their abundance fitness scores by subtracting their abundance error (Sigma$_{scaled}$; Eq. 6). Variants passing both criteria are colored green in Fig. 3c.

To assess other variants at functionally important positions we further gated this plot as indicated in Supplementary Fig. 20, grouping remaining variants into 5 categories: low abundance, but active; abundant and active (wildtype-like); partially abundant and partially active; abundant, but partially active; and low abundance and inactive. The result of this extended analysis is reported in Supplementary Table 3.

### Identification of 3k and 5c drug escape hotspots
We first attempted to extract drug-escape variants from a plot of drug escape scores versus activity scores, however in follow up experiments we found our dataset was confounded by poor dox control of some

variants. We instead identified drug-escape variants by plotting drug-escape scores versus leaky expression scores. We noted disperse populations for synonymous wildtype and nonsense variants in these plots and performed additional cleanup of the data (Supplementary Fig. 12). Variants defined with less than 5 barcodes were discarded as were rare variants with much higher errors than the main population. When these filters were applied to synonymous wildtype variants the population dispersion was considerably narrowed. We used surviving synonymous wildtype variants to determine a final minimum counts filter that was applied to the whole dataset.

The filtered data was then re-plotted (drug-escape vs leaky expression) to identify drug-escape variants that were defined as follows. We drew a density border around synonymous wildtype variants using geom_density2d from ggplot2 showing the outermost contour. At the top and bottom borders of this contour we extended the linear relationship between drug-escape and leaky expression (Fig. 4c). Variants that had drug-scape scores greater than the right contour boundary of the synonymous wildtype variants and were below the bottom linear correlation line were considered true escape variants.

### Recombinant PLpro expression and purification

Purification of recombinant PLpro was adapted from Calleja et al. [21]. Wildtype PLpro and variants were transformed into BL21/DE3 (NEB, #C2527I) and grown in Super Broth (WEHI Bioservices) with 50 µg/ml Kanamycin (Gibco, #11815032). Once an optical density of 0.8 was reached, the temperature was dropped to 18 °C and expression was induced with 300 µM IPTG (Bioline, #BIO-37036). Cells were harvested 16h-18h after induction and pelleted. Pellets were resuspended in lysis buffer (50 mM Tris.HCl (Invitrogen, #15506017) pH 7.5, 500 mM NaCl (Supelco, #1.93206), 10 mM Imidazole (Sigma Aldrich, #I202) pH 8.0, 5 mM beta-mercaptoethanol (Merck, #M6250-100ML), freshly supplemented with lysozyme (Sigma Aldrich, #L6876), DNaseI (Roche, #10104159001) and cOmplete protease inhibitor cocktail (Roche, #11873580001)) and sonicated. Supernatants after high-speed centrifugation were loaded onto His-Tag Purific Resin (Roche, #05893682001), washed and eluted with 50 mM Tris.HCl pH 7.5, 500 mM NaCl, 300 mM Imidazole pH 8.0, 5 mM beta-mercaptoethanol. The eluate was desalted with a PD-10 column (Cytiva #17-0851-01) into 50 mM Tris.HCl pH 7.5, 500 mM NaCl, 10 mM Imidazole pH 8.0, 5 mM beta-mercaptoethanol and cleaved with 3 C protease (made in-house) overnight. The next day, protein was again passed over Nickel resin to remove His-tagged protein and further purified by size exclusion chromatography (Superdex 75 10/300 g, GE healthcare) into 20 mM Tris.HCl 7.5, 50 mM NaCl, 1 mM TCEP (Merck, #646547-10X1ML). Fractions were analyzed with SDS-PAGE and correct fractions were pooled, concentrated, aliquoted, flash-frozen and stored at −80 °C.

### Recombinant PLpro activity assays

We used the PLpro substrate Z-RLRGG-AMC acetate (Sigma Aldrich, #SML2966) for inhibitor dose response assays. Inhibitors were pre-diluted in DMSO at 50-fold their final concentrations and tested in a 10-point, 3-fold dilution series with 100 µM as the top final concentration. 120 nL of inhibitor was first spotted in wells (Echo® acoustic dispenser, LabCyte) of a 384-well black plate (Corning #3820) and PLpro and Z-RLRGG-AMC added to initiate the reaction (6 µl of 10 nM PLpro and 300 nM Z-RLRGG-AMC in 50 mM HEPES (Gibco, #11344041) pH 7.5, 0.1 mg/ml bovine serum albumin (Sigma Aldrich, #A7030), 150 mM NaCl, 2.5 mM dithiothreitol (Invitrogen, #15508013)). Dithiothreitol was added to allow cysteine regeneration during the catalytic cycle. The mixture was incubated at room temperature for two hours. AMC fluorescence with excitation at 380 nm and emission of 445 nm was measured on a ClarioStar Plus (BMG labtech).

For $k_{cat}/K_M$ measurements, an 8-point, 3-fold serial dilution of Z-RLRGG-AMC starting at 25 µM was incubated with 5 nM PLpro in the

same assay buffer as above (reaction volume 6 µl). To measure the fluorescence of completely cleaved substrate we incubated the same Z-RLRGG-AMC dilution series with 100 nM PLpro. Base measurements of the dilution series in the absence of PLpro were also taken to measure the fluorescence from intact substrate. The reaction was scanned every 5 min for 2 h to measure initial rates. The same strategy was used for Ubiquitin-Rhodamine110 (UbiQ, UbiQ-002) and ISG15-Rhodamine110 (UbiQ, UbiQ-127) substrates with the following alterations. The top concentration in the dilution series was 500 nM, the assay buffer was 20 mM Tris pH 8.0, 0.03% BSA, 0.01% Triton X-100 (Merck, #X100-6X500ML), 1 mM L-glutathione (Sigma, #G4251) and fluorescence was measured at an excitation of 487 nm and an emission of 535 nm. 5 nM PLpro was used in the Ub-Rhodamine110 assay while 100 nM PLpro was used to capture the fluorescence of complete cleavage; 0.5 nM PLpro was used in the ISG15-Rhodamine110 assay while 10 nM PLpro for ISG15-Rhomdaine was sufficient for complete cleavage in 2 h. As the substrate concentrations used in these assays were well below the estimated $K_M$ for each substrate, we approximated the $k_{cat}/K_M$ from the slope of plots of initial rate/PLpro concentration vs substrate concentration with linear regression (Eq. 7). A similar approach has been used previously[13]

$$\frac{[k_{cat}]}{[K_M]} \approx \frac{[V]}{[S][PLpro]} \qquad (7)$$

Equation 8 is the conventional Michaelis-Menten equation:

$$[V] = \frac{[k_{cat}]*[PLpro]*[S]}{[K_M] + [S]} \qquad (8)$$

As $[K_m] \gg [S]$, the denominator was approximated to $[K_m]$.

### Thermal shift assay

20 nM PLpro in 20 mM Tris-HCl, 50 mM NaCl, 1 mM TCEP with 8X SYPRO Orange dye (Supelco, #S5692) were filtered in SpinX tubes (Corning, #CLS8160) and 25 µl added to a 384-well PCR plate (Thermo, AB-1384/W). Samples were heated for 30 s at the temperature starting at 25 °C to 95 °C by 1 °C increment and SYPRO Orange fluorescence monitored with a Biorad CFX384 Real Time System, C1000 Thermal Cycler. Data were normalized with minimum fluorescence dictating 0 and maximum fluorescence as 100. Melting temperatures (V50) were extracted with regression using Eq. 9 in Prism 9:

$$Y = Bottom + \frac{(Top - Bottom)}{1 + e^{\frac{(V50-X)}{Slope}}} \qquad (9)$$

### M208W PLpro structure determination

PLpro M208W protein was concentrated to 10 mg/ml and sent to the WEHI-Bio21-Crystallisation Facility for SG1 (shotgun) screening in a vapor diffusion sitting drop plate (96-well plate, 100 nL reservoir buffer + 100 nL protein). The best condition was identified as 0.1 M trisodium citrate pH 5.5 + 20% w/v PEG3000. Crystals from this well were harvested and made into seeds with Hampton Research's Seed Bead Kit and used in a hanging drop setup with 1 µl 10 mg/ml M208W PLpro plus 1 µl mother liquor (0.1 M trisodium citrate (Univar, #AJA467-500) pH 5.5, 20% w/v PEG3000 (Merck, #81227)) and 0.2 µl seeds. Crystals from this drop were fished and cryoprotected with 0.1 M trisodium citrate pH 5.5, 20% w/v PEG3000, 30% glycerol before vitrification. Data were collected from the Australian Synchrotron (Australian Nuclear Science and Technology Organization, ANSTO) beamline, MX2[66] at a wavelength of 0.9537 Å and a temperature of 100 K. Data were indexed with XDS[67] and scaled with Aimless[68] and processed with Truncate[69]. The 2.00 Å structure of M208W PLpro (8VEC) was solved using Phaser[70] by molecular replacement with 8FWN. R-free flags were

selected in Phenix[71]. The refinement in PHENIX started with a round of rigid body refinement followed by simulated annealing. Subsequently, each refinement cycle consisted of XYZ (reciprocal, real-space) and individual B-factor refinement, and model building in Coot[72]. Zinc was positioned in the density center near the zinc-finger and water molecules were added in the final stages of refinement. The refinement statistics have been summarized (Supplementary Table 1) and the omit map density around Trp208 has been provided (Supplementary Fig. 17).

### B-factor comparison

We compared the B-factors of C111S (7D47) and M208W (8VEC) against 53 other PLpro structures in the PDB with resolution lower than 3.5 Å to look for regions that are stabilized by the respective mutations. Because crystallization conditions and data quality alter B-factors in ways that are not intrinsic to the protein of interest we scaled each PDB's B-factors to either C111S or M208W data before looking for regions of altered stability when compared to the population.

Tables were made for each PDB file and contained columns for the residue position and its average B-factor calculated from C, Cα, N, O and, when not Glycine, Cβ atoms. The Zinc finger residues 186 to 197 and 219 to 232 were excluded in the analysis as they are disordered in many PLpro structures. The true intrinsic B-factors ($B_{true\_resi\_N}$) of PLpro were estimated from the average of all 55 PDB files, which had multiple spacegroups and crystallization conditions.

To determine regions of stabilization and destabilization in M208W (8VEC) and C111S (7D47) we used the "optim" function in R (R studio v4.2.0), using the Broyden–Fletcher–Goldfarb–Shanno (BFGS) method[46] to find scaling (k) and translating (b) constants and rescaled B-factors according to Eq. (10).

$$B_{transformed\_resi\_N} = k \times B_{[PDB]\_resi\_N} + b \qquad (10)$$

The BFGS method is an iterative process that we have used to converge on appropriate scaling values by minimizing the sum-of-squares total differences between B-factor tables. The sum-of-squares total differences formula is defined by Eq. (11).

$$Total_{difference} = \sum \left( \frac{B_{transformed\_resi\_N} - B_{[M208W\ or\ C111S]\_resi\_N}}{B_{true\_resi\_N}} \right)^2 \qquad (11)$$

where $B_{transformed\_resi\_N}$ is the average residue B-factor at position N of the PDB file upon applying transformation, $B_{[M208W\ or\ C111S]\_resi\_N}$ is the average residue B-factor at position N of the PDB file being assessed, while $B_{true\_resi\_N}$ is the average residue B-factor at position N from the average of all PDB files.

The success of this approach is dependent on the initial values of k and b, which are defined in Eq. 12. k was specified as the average of all residue B-factors of M208W or C111S divided by the average of all residue B-factors of the PDB structure of interest and b was initiated from zero:

$$k_{initial} = \frac{B_{[M208W\ or\ C111S]\_avg}}{B_{[PDB]\_avg}}; b_{initial} = 0 \qquad (12)$$

where $B_{[M208W\ or\ C111S]\_avg}$ is the average B-factor of M208W or C111S; $B_{[PDB]\_avg}$ is the average B-factor of the PDB file without applying any transformation.

Once optimal k and b values were determined we plotted data from all scaled PDB B-factors to ensure scaling was successful (Fig. 5d). We also applied the same normalization parameters to the Wilson B-factor for each structure (Supplementary Table. 2). The transformed Wilson B-factors are within 5Å² of the Wilson B-factor of our M208W structure, showing that our transformation is appropriate. Residues from M208W or C111S PLpro with B-factors more than 2 standard deviations higher than the mean of all other data were reported as destabilized and those more than 2 standard deviations below were classified as stabilized residues.

### PLpro sequence alignment

The alignment of SARS-CoV-1, HKU1, 229E, NL63, OC43, BtSARS, Bt273, Bt133, BtHKU9, MHVJ, BCoV, TGEV, and aIBV PLpro was taken from Chaudhuri et al.[36], who considered structural elements during their alignment. We took the alignment between MERS PLpro, SARS-CoV PLpro and SARS-CoV-2 from Klemm et al.[6] that also considered the structural alignment and manually merged them based on the SARS-CoV PLpro sequence, which was used to generate a WebLogo representation[73] in Fig. 2b. The alignment of all sequences can be found in Supplementary Data 3.

### Identification of circulating variants

We obtained circulating variants from the online resource COVID-3D[35]. We first examined the variants in the Nsp3 PLpro region (aa746 to 1060) and found the variant with the maximum observations, which is A145D with 24688 sequences registered. In our analysis we only considered variants that had observations of more than 1% of the most observed variant (i.e 247) to exclude rare variants that might be due to sequencing errors.

### Reporting summary

Further information on research design is available in the Nature Portfolio Reporting Summary linked to this article.

## Data availability

The DMS data generated in this study have been deposited in the MaveDB database under accession code mavedb:00000672. The crystallography data generated in this study have been deposited in the PDB under accession code 8VEC. The data used to generate figures in this study are provided in the Source Data file. The PDB files used in this study are available in the PDB and are listed in Supplementary Table 2. Source data are provided with this paper.

## Code availability

Code used in this study is available in Code Ocean (https://doi.org/10.24433/CO.0426220.v1).

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

## Acknowledgements

This work has been supported by: a Wellcome Trust Innovator Award 222698/Z/21/Z to D.K., G.L., and M.J.C., WEHI Innovator funding to D.K., G.L., and M.J.C., MRFF grants MRF2002119 to D.K. and G.L., and MRF2016781 to D.K., G.L., and M.J.C.; NHMRC Investigator Grants (GNT2016461 to GL, GNT1178122 to DK); and a donation from John and Tibby Peterson to M.J.C. This research was undertaken in part using the MX2 beamline at the Australian Synchrotron, part of ANSTO, and made use of the Australian Cancer Research Foundation (ACRF) detector. The D.M.S. libraries were created with the help of the WEHI Multiplexed Assay Technology Hub, which was founded with WEHI New Medicines and Advanced Technology Theme funding. We would like to thank the Bio21-WEHI Crystallisation Facility within Melbourne Protein Characterisation at The Bio21 Molecular Science and Bio-technology Institute, The University of Melbourne. We thank WEHI Flow Cytometry and Advanced Genomics Facilities as well as the National Drug Discovery Centre for technical resources and advice. Work in the laboratories of the authors was made possible through Victorian State Government Operational Infrastructure Support (OIS); Australian Government NHMRC Independent Research Institute Infrastructure Support (IRIIS) Scheme; a donation from Hengyi Pacific Pty Ltd to support COVID-19 research; and a donation from AWM Electrical to support Australian drug discovery research to the National Drug Discovery Centre.

## Author contributions

M.J.C. conceived and with M.E.C. and D.K. supervised the study. X.W. and M.G. designed the libraries. M.J.C. and X.W. designed the cell assays and D.M.S. strategies. X.W., M.G. prepared the samples and carried out the Illumina sequencing. X.W. and M.G. prepared the samples for PacBio sequencing. All data were processed and analyzed by X.W., under the guidance from M.J.C. and M.E.C. X.W. prepared samples for Oxford Nanopore sequencing and developed the pipeline to process the data, while K.Z. provided critical guidance in Nanopore data processing. X.W. performed all cell experiments, and with B.G.C.L. performed the bio-chemical assays. X.W. purified all proteins and with J.V.N. performed the crystallization study. D.J.C. provided advice on PLpro protein purifica-tion and crystallization. M.J.C. and X.W. processed and analyzed the crystallization data. D.K., G.L., J.P.M., B.G.C.L., K.N.L. and N.W.K. pro-vided PLpro inhibitors and developed biochemical assays used in this study. X.W. and M.J.C. drafted the manuscript, which was reviewed and edited by all co-authors.

## Competing interests

DK is founder, shareholder and Scientific Advisory Board member of Entact Bio. The remaining authors declare no competing interests.
