## [Peer Review File · Nature Communications]

Mutational Profiling of SARS-CoV-2 Papain-Like protease reveals requirements for function, structure, and drug escape.REVIEWER COMMENTS

Reviewer #1 (Remarks to the Author):

Wu et al. present a series of deep mutational scanning experiments on SARS-CoV-2's PLpro papain-like protease. They nicely present the experiments that demonstrate the utility of the cell-based fluorescence biosensor PLPro activity assay and its suitability for mutational scanning. Mutational scanning results support the constraint and tolerability to mutation of sites that is concordant with structural and evolutionary reasoning while nonetheless pointing out new trends (functional constraint that extends out from the direct substrate binding pocket toward the fingers). They extend the method toward the extremely important purpose of screening resistance pathways against PLpro-targeting drugs currently under active investigation together with validation experiments. Last, they investigate mutations at one curious site at an intramolecular domain interface (M208) through biophysical and structural analysis, suggesting that mutations at this site can increase or decrease activity based on how they impact flexibility of the protein at the boundary of these two domains.

Overall, this is a quality and important study. I have two 'more important' suggestions (I wouldn't call them "major" though), and a couple of minor comments.

1. It's unclear if and how replicates were performed anywhere in the main text, methods, or figures. In the methods, line 801 describes 'experimental replicates were defined as selections from independent transductions', suggesting replicate library preparation and FACS-seq experiments were performed. If so, correlation plots between mutant phenotypes (as well as average mutational effect per site, since many trends are more about the overall tolerability of a site) from experimental replicates must be included to help the reader gauge how precise measurements are at the per-mutation and per-site level, and the main text should describe how replicates were performed (e.g., separate transductions of the same barcode-plasmid pool that went through separate FACS binning; versus replicate FACS binning of the same transduced cell population). If true replicates were not performed, I would strongly suggest that a second replicate assay be run and included.

2. Normalized range being -1 to 1 is very odd. I know a -1 (or -Inf) to 0 scale is also arbitrary, but is much more standard, since everybody is used to 0 'meaning something' (i.e., then a positive number means improvement and negative number decrease), whereas in this -1 to 1 scale, the value of 0 is meaningless. Having an arbitrary zero value be the white scale in a diverging color scale bar for a heatmap is also extremely confusing. If the midpoint of a color scale doesn't 'mean something', then one should just use a single tone color scale (i.e. white to blue or white to red, not two diverging colors in different direction from a meaningless reference number). If setting the white point to be the mean wildtype measurement creates too much shading in the heatmaps that the authors think is misleading (i.e. mutations looking 'worse than wildtype' due to noise), in ggplot2 in R one can modify the boundaries of a diverging color scale using the `scale_fill_gradientn` option, for example placing the "white" color at multiple value points along a color scale. One sensible thing to do would be, for example, to compute the 20th and 80th percentile (or whatever number gives an appropriate visualization) of functional scores for the redundant synonymous controls, and define that window of phenotypes as indistinguishable from wildtype therefore to be colored white. Alternatively, a single color scale here would also be appropriate and give less emphasis to possible noise.

Minor:

“highly homologous” is improper term – something is either homologous (evolutionarily related) or not, there is no quantitative scale to homology.

It would be helpful if Extended Data Fig. 3, or at the very least the Methods, explained how “error” was determined

Fig. 3-5 mislabeled as Fig. 1-3 in caption?

Reviewer #2 (Remarks to the Author):

The manuscript presents an impressive body of work on SARS-CoV-2 papain-like protease (PLpro), an enzyme essential for viral replication. It applies deep mutational scanning (DMS) technology in mammalian cells to evaluate mutational effects on the stability and activity of PLpro, and to identify potential escape variants from two prominent PLpro inhibitors, 3k and 5c. These DMS data informed variant selection for recombinant protein expression and activity assays, which largely confirm the DMS predictions. This study highlights the power of DMS in not only to understand expression, activity and allostery but also to identify mutations that may provide escape from inhibitors. An interesting allosteric variant, M208W, was crystallized, and its structural properties discussed. The data are well-curated, the figures are clear and the motivations for the work are topical and coherent.

Point-by-Point

Line 125: doxycycline (dox) is introduced. Suggesting adding... introduced to drive PLpro expression.

Lines 220-222: ~20% loss of variant data is a high proportion (and causes the heatmap in Extended Fig. 4a to be very patchy). Can you add a sentence or two explaining what you think the loss of redundancy is caused by? Are your filtering steps too stringent?

Lines 702-736: PLpro library construction and PLpro Abundance library construction.

Different cloning strategies are used to construct the library at different sites. For the first library, dsDNA with gateway adapters are used for residues close to the active site (residues 62,69-70, 73-74, 77, 93, 104, 106-119, 151-174, 206-212, 243-253, 260-276, 285-286, 296-304) and MITE mutagenesis for the remaining residues. I wonder if this will affect relative abundances when it comes to PacBio characterization. Would it be possible to provide an extended figure showing relative abundances of variants, differentiated according to whether they were or were not one of the residues for which dsDNA with gateway adapters were ordered? Likewise, for the abundance library construction, it is stated that only residues up to L290 are retained, and so new variants between L290 and K315 are created with another round of MITE mutagenesis. Would it be possible to provide an extended figure showing relative abundances of variants, differentiated according to whether they are or are not one of these newly-created variants?

Lines 1414-1424. Extended data figure 1. Why would reduced FRET result in a drop in mClover3 fluorescence? I think this needs to be addressed further, the authors could at least

give some proposed reasons for why this may be the case. It would be good to address this both in the caption to this figure, and in the main text (somewhere between lines 113 and 133).

Extended Figures 6, 7 and 9. Variants at the C-terminus (~309-315, esp 314 and 315) are shown to typically have high fitness scores, yet this is never addressed in the text (whether this is a real effect or an experimental artifact). I request the authors add a sentence or two to explain their thoughts.

Minor issues:

Lines 86-87 and 98. Avoid saying that the study assesses the impact of all single-site substitutions, as this is not quite true. Instead, use 'vast majority' or something similar.

Line 155. 'Ensure' is too strong a term.

Figures 3 (line 1326), 4 (line 1348) and 5 (line 1387) are incorrectly labelled as Figures 1, 2 and 3.

Line 263. Typo, hyphen not needed.

Line 375. Typo, 'comparable' not 'compared'.

Line 446. Typo, should be 'not even Asn', rather than 'not even Asp'.

Line 666. Typo, 'subsequently', not 'subsequentially'.

Line 677. Typo, 'MERS', not 'MERs'.

Line 778. Typo, 'at to'.

Lines 803-806. Restructure this sentence because as it is written, it is implied that Fav_g-syn equals 1 and Fav_g-nonsense(res1-305) equals -1, which is misleading. I would recommend rewriting as: 'Instead, we wanted to normalize data to center the population of synonymous wildtype variants (Fav_g-syn) to a score of 1 and the population of nonsense variants (Fav_g-nonsense(res1-305)) to a score of -1 for easy visualization of variants and comparison of datasets.'

Line 923. Typo, 'with for'.

Line 952. Typo, 'with using'.

Line 983. Where is the * which this footnote refers to?

Reviewer #3 (Remarks to the Author):

We have carefully read the reviewer comments, thank them for their feedback and respond to the points made by reviewers below.

Reviewer #1

1. It's unclear if and how replicates were performed anywhere in the main text, methods, or figures. In the methods, line 801 describes 'experimental replicates were defined as selections from independent transductions', suggesting replicate library preparation and FACS-seq experiments were performed. If so, correlation plots between mutant phenotypes (as well as average mutational effect per site, since many trends are more about the overall tolerability of a site) from experimental replicates must be included to help the reader gauge how precise measurements are at the per-mutation and per-site level, and the main text should describe how replicates were performed (e.g., separate transductions of the same barcode-plasmid pool that went through separate FACS binning; versus replicate FACS binning of the same transduced cell population). If true replicates were not performed, I would strongly suggest that a second replicate assay be run and included.

Thank you for pointing out that we were missing detail on how replicates were performed. This information was in a previous version of the manuscript but was lost when we cut down text to conform to manuscript word limits. We have updated the main text to clarify the number and nature of replicates. We performed between 2 and 7 replicates for all our datasets. All our replicates were independent transductions that were never mixed or resampled. In some instances, independent transductions were performed from the same virus pool.

To provide correlation plots at the individual variant level for all replicate scores would require 308,700 plots for the activity dataset alone. Instead, we have provided single graphs for DMS activity and abundance datasets that plot replicates by variant. Individual scores for each replicate are also provided in table format in MaveDB. Additionally, the overall precision of each screen can be judged by the histograms supplied with each sequence-function map where variants that produce wildtype PLpro are coloured in red, and variants that encode truncated PLpro products are coloured in blue. The segregation and spread of both synonymous wildtype and nonsense variants are indicative of the power of the screen.

2. Normalized range being -1 to 1 is very odd. I know a -1 (or -Inf) to 0 scale is also arbitrary, but is much more standard, since everybody is used to 0 'meaning something' (i.e., then a positive number means improvement and negative number decrease), whereas in this -1 to 1 scale, the value of 0 is meaningless. Having an arbitrary zero value be the white scale in a diverging color scale bar for a heatmap is also extremely confusing. If the midpoint of a color scale doesn't 'mean something', then one should just a single tone color scale (i.e. white to blue or white to red, not two diverging colors in different direction from a meaningless reference number). If setting the white point to be the mean wildtype measurement creates too much shading in the heatmaps that the authors think is misleading (i.e. mutations looking 'worse than wildtype' due to noise), in ggplot2 in R one can modify the boundaries of a diverging color scale using the `scale_fill_gradientn` option, for example placing the "white" color at multiple value points along a color scale. One sensible thing to do would be, for example, to compute the 20th and 80th percentile (or whatever number gives an appropriate visualization) of functional scores for the redundant synonymous controls, and define that window of phenotypes as indistinguishable from wildtype therefore to be colored white.

Alternatively, a single color scale here would also be appropriate and give less emphasis to possible noise.

We concur that normalising from -1 to 1 is non-standard and have updated our paper to scale between 0 and 1, where 0 is indicative of no PLpro activity and 1 represents wildtype levels of PLpro activity. After trialling a number of colour schemes, including a single colour scale, we respectfully feel that the original colour scheme is the most informative. Our data is close to being bimodal, where most variants are either fully active or fully impaired. The diverging colour scheme still allows one to see which variants have intermediate activity (white is the midpoint) and light blue and red shading allows one to determine if an impaired variant is closer to inactive or active variants. This is hard to do with a single colour gradient. We have included a bar under the histogram plots with each dataset so that the reader can interpret the range of shades occupied by synonymous wildtype and nonsense variants without the need to hide noise in the data. A similar colour scheme has been used to present SARS-CoV-2 Mpro DMS data by an independent group¹ and we feel that this is the most informative when assessing protein function by DMS.

¹Flynn JM, Samant N, Schneider-Nachum G, Barkan DT, Yilmaz NK, Schiffer CA, Moquin SA, Dovala D, Bolon DNA. Comprehensive fitness landscape of SARS-CoV-2 M^{pro} reveals insights into viral resistance mechanisms. *Elife*. 2022 Jun 20;11:e77433. doi: 10.7554/eLife.77433.

“highly homologous” is improper term – something is either homologous (evolutionarily related) or not, there is no quantitative scale to homology.

Agreed. The text has been updated to "evolutionarily related" as suggested.

It would be helpful if Extended Data Fig. 3, or at the very least the Methods, explained how “error” was determined

We have updated the methods to clarify that errors were calculated in DiMSum. DiMSum provides comprehensive error modelling that considers that single protein variants can be encoded by multiple codons, technical replicates are introduced during sequencing, and errors vary depending on the abundance of a particular variant in the library. More information on DiMSum's treatment of errors in a DMS experiment can be found in this publication.

Faure, A.J., Schmiedel, J.M., Baeza-Centurion, P. *et al.* DiMSum: an error model and pipeline for analyzing deep mutational scanning data and diagnosing common experimental pathologies. *Genome Biol* **21**, 207 (2020). <https://doi.org/10.1186/s13059-020-02091-3>

Fig. 3-5 mislabeled as Fig. 1-3 in caption?

We apologise for this oversight and have updated the caption legends.

Reviewer #2&3:

Line 125: doxycycline (dox) is introduced. Suggesting adding... introduced to drive PLpro expression.

We have updated the passage in question to clarify the link between dox and PLpro expression:

"To validate the biosensor's ability to report on PLpro activity, we installed the PLpro coding sequence into a lentiviral Tet-On expression vector so PLpro expression could be driven by doxycycline (dox) addition. Upon introduction of this construct into our 293T biosensor cell line, FRET emission was reduced in a dox dependent manner, indicating PLpro dependent biosensor cleavage (Fig. **1b-e**)."

Lines 220-222: ~20% loss of variant data is a high proportion (and causes the heatmap in Extended Fig. 4a to be very patchy). Can you add a sentence or two explaining what you think the loss of redundancy is caused by? Are your filtering steps too stringent?

The original abundance library data was patchy because we underestimated the number of colonies required to achieve full coverage when reformatting the library. We went ahead regardless after investing in PacBio sequencing, and we felt valuable data could be gleaned regardless. Nevertheless, while the paper was in review, we recloned our abundance library and trialled a new Nanopore-based long-read sequencing pipeline, spiking in our PacBio validated library as a measure of Nanopore fidelity. With excellent agreement between Nanopore and PacBio long-read sequencing (99% of variants) we rescreened for PLpro abundance and now have 97% coverage. With the new data we have updated Table 1 and made minor changes in the discussion related to the new mutations identified as functional variants.

Lines 702-736: PLpro library construction and PLpro Abundance library construction. Different cloning strategies are used to construct the library at different sites. For the first library, dsDNA with gateway adapters are used for residues close to the active site (residues 62,69-70, 73-74, 77, 93, 104, 106-119, 151-174, 206-212, 243-253, 260-276, 285-286, 296-304) and MITE mutagenesis for the remaining residues. I wonder if this will affect relative abundances when it comes to PacBio characterization. Would it be possible to provide an extended figure showing relative abundances of variants, differentiated according to whether they were or were not one of the residues for which dsDNA with gateway adapters were ordered? Likewise, for the abundance library construction, it is stated that only residues up to L290 are retained, and so new variants between L290 and K315 are created with another round of MITE mutagenesis. Would it be possible to provide an extended figure showing relative abundances of variants, differentiated according to whether they are or are not one of these newly-created variants?

We have included new supplemental figures 18 and 19 that show the overall counts of our libraries across all variants and a comparison of the positions cloned by Twist synthesized DNA fragments and MITE mutagenesis.

Lines 1414-1424. Extended data figure 1. Why would reduced FRET result in a drop in mClover3 fluorescence? I think this needs to be addressed further, the authors could at least give some proposed reasons for why this may be the case. It would be good to address this both in the caption to this figure, and in

the main text (somewhere between lines 113 and 133).

We also noticed this, but as both the drop in mClover3 fluorescence and reduced FRET are PLpro dependent processes it did not impact our analysis and we did not investigate the mechanism further. We can think of three reasons that could account for this phenomenon. The appendage of TLKGG-COOH to the end of mClover3 destabilizes mClover3; there is a cryptic PLpro cleavage site in mClover3 itself; or that mRuby3 has a stabilizing effect on mClover3 when fused. We have adjusted the main text and supplementary figure legend with the following statement, which acknowledges this quirk without being overly speculative:

"Fluorescence in the donor channel unexpectedly dropped upon biosensor cleavage. The reduction of mClover3 fluorescence indicates that free mClover3 fused to a fragment of the PLpro cleavage motif (T_{P5}L_{P4}K_{P3}G_{P2}G_{P1}) is less stable than the intact biosensor. We therefore present data as FRET versus acceptor fluorescence."

Extended Figures 6, 7 and 9. Variants at the C-terminus (~309-315, esp 314 and 315) are shown to typically have high fitness scores, yet this is never addressed in the text (whether this is a real effect or an experimental artifact). I request the authors add a sentence or two to explain their thoughts.

The following text has been added to the manuscript in Results: *Predicting drug-escape variants with DMS*:

We plotted leaky expression fitness scores and observed a non-random distribution of positions in the library with poor dox control suggesting protein intrinsic properties were the basis of this phenomena. Particularly obvious were aromatic residues at positions 170 and 208, which both point into the cleft between the thumb and palm domains, as well residues 313-315 at the C-terminus that are distal from the active site. Since PLpro resides within the multidomain NSP3 protein, the C-terminus is not normally exposed. We suspect that the last three residues of the PLpro construct are somewhat destabilizing and mutation or truncation reduces PLpro turnover in mammalian cells.

Lines 86-87 and 98. Avoid saying that the study assesses the impact of all single-site substitutions, as this is not quite true. Instead, use 'vast majority' or something similar.

We agree and have altered our language accordingly.

Line 155. 'Ensure' is too strong a term.

We agree and have changed "...to ensure that cells..." to "...so that most cells..."

Figures 3 (line 1326), 4 (line 1348) and 5 (line 1387) are incorrectly labelled as Figures 1, 2 and 3.

We apologise for this oversight and have updated the caption legends.

Line 263. Typo, hyphen not needed.

Agreed. The hyphen has been removed from drug-treatment.

Line 375. Typo, 'comparable' not 'compared'.

We have clarified the intended meaning of this sentence:

"After minimal optimization of screening conditions, we collected a 2.00 Å dataset of M208W with **unique** space group and unit cell dimensions (P2₁2₁2; unit cell 58 Å, 85.5 Å, 83 Å, 90°, 90°, 90°) (Supplementary Table 1) that were unique compared to other PLpro structures in the PDB."

Line 446. Typo, should be 'not even Asn', rather than 'not even Asp'.

Line 666. Typo, 'subsequently', not 'subsequentially'.

Line 677. Typo, 'MERS', not 'MERs'.

Line 778. Typo, 'at to'.

Line 923. Typo, 'with for'.

Line 952. Typo, 'with using'.

Agreed, these typos have been updated in the text.

Lines 803-806. Restructure this sentence because as it is written, it is implied that Fav_{g-syn} equals 1 and Fav_{g-nonsense(res1-305)} equals -1, which is misleading. I would recommend rewriting as: 'Instead, we wanted to normalize data to center the population of synonymous wildtype variants (Fav_{g-syn}) to a score of 1 and the population of nonsense variants (Fav_{g-nonsense(res1-305)}) to a score of -1 for easy visualization of variants and comparison of datasets.'

In this normalization we used actual wildtype rather than synonymous wildtype. We have clarified this sentence to the below text:

"Instead, we wanted to normalize data to center the scores from barcodes linked to wildtype PLpro to a score of 1 (F_{WT}) and the population of nonsense variants to a score of 0 (F_{avg-nonsense(res1-304)}) for easy visualization of variants and comparison of datasets."

*Line 983. Where is the * which this footnote refers to?*

The footnote has been deleted from its current position and moved into the method text.

REVIEWERS' COMMENTS

Reviewer #1 (Remarks to the Author):

My original comments have been replied to and addressed as needed, and I remain supportive of the manuscript and the work.

Reviewer #2 (Remarks to the Author):

The authors have address the relevant critiques and the article is now suitable for publication.

Reviewer #3 (Remarks to the Author):

Dear Authors,

Thank you for addressing the comments made during the first round of review. I am happy with all of the alterations made, and the new figures introduced in the supplementary material. I am especially encouraged by the new experimental material using Nanopore-based long-read sequencing. This has improved library coverage from ~80% to 97%, and agreement between this and the previous PacBio sequencing appears to be excellent, allowing for a fuller analysis of functional variants.

The following statement, from the first round of review, still applies:

The manuscript presents an impressive body of work on SARS-CoV-2 papain-like protease (PLpro), an enzyme essential for viral replication. It applies deep mutational scanning (DMS) technology in mammalian cells to evaluate mutational effects on the stability and activity of PLpro, and to identify potential escape variants from two prominent PLpro inhibitors, 3k and 5c. These DMS data informed variant selection for recombinant protein expression and activity assays, which largely confirm the DMS predictions. This study highlights the power of DMS in not only to understand expression, activity and allostery but also to identify mutations that may provide escape from inhibitors. An interesting allosteric variant, M208W, was crystallized, and its structural properties discussed. The data are well-curated, the figures are clear and the motivations for the work are topical and coherent.